# Pathobiology of Highly Pathogenic Avian Influenza A (H5N1 Clade 2.3.4.4b) Virus from Pinnipeds on Tyuleniy Island in the Sea of Okhotsk, Russia

**DOI:** 10.3390/v18010051

**Published:** 2025-12-29

**Authors:** Alexander Alekseev, Ivan Sobolev, Kirill Sharshov, Marina Gulyaeva, Olga Kurskaya, Nikita Kasianov, Maria Chistyaeva, Alexander Ivanov, Olesia Ohlopkova, Aleksey Moshkin, Marina Stepanyuk, Anastasiya Derko, Mariya Solomatina, Batyrgishi Mutashev, Mariya Dolgopolova, Alimurad Gadzhiev, Alexander Shestopalov

**Affiliations:** 1Federal Research Center of Fundamental and Translational Medicine, 630060 Novosibirsk, Russia; sobolev.riov@yandex.ru (I.S.); sharshov@yandex.ru (K.S.); mgulyaeva@gmail.com (M.G.); kurskaya_og@mail.ru (O.K.); nskasianov@frcftm.ru (N.K.); ohlopkova.lesia@yandex.ru (O.O.); alex.moshkin727@gmail.com (A.M.); stepanunya1996@gmail.com (M.S.); a.derko19@gmail.com (A.D.); solomatina.mariyav@yandex.ru (M.S.); mutashev2001@bk.ru (B.M.); m.dolgopolova.nsu@gmail.com (M.D.); shestopalov2@mail.ru (A.S.); 2Faculty of Ecology and Sustainable Development, Dagestan State University, 367016 Makhachkala, Russia; ali-eco@mail.ru; 3Department of Natural Sciences, Novosibirsk State University, 630090 Novosibirsk, Russia; 4Autonomous Non-Commercial Organization “Tourist and Ecological Club “Boomerang”, 693021 Yuzhno-Sakhalinsk, Russia; vamp_lestat@mail.ru; 5Green Sakhalin Nature and Environment Protection Fund, 694620 Kholmsk, Russia; greensakhalin@yandex.ru

**Keywords:** highly pathogenic avian influenza, H5N1, clade 2.3.4.4b, northern fur seal, pinnipeds, Tyuleniy Island, Sea of Okhotsk, interspecies transmission, pathogenicity, phylogenetics

## Abstract

Highly pathogenic avian influenza (HPAI) A(H5N1) clade 2.3.4.4b has recently emerged as a major threat to wildlife, agriculture, and public health due to its expanding host range and the increasing frequency of spillover into mammals. In July–August 2023, the mass death of over 3500 northern fur seals (*Callorhinus ursinus*) and at least one Steller sea lion (*Eumetopias jubatus*) was recorded on Tyuleniy Island in the Sea of Okhotsk, Russia. Two HPAI A(H5N1) viruses were isolated from fur seal carcasses and designated A/Northern_fur_seal/Russia_Tyuleniy/74/2023 and A/Northern_fur_seal/Russia_Tyuleniy/75/2023. Both viruses exhibited high pathogenicity in chickens (IVPI 2.7–3.0) and mice (MLD_50_ 1.9–2.5 log_10_EID_50_/mL), with distinct differences in disease progression, histopathology, and organ tropism. Experimental infection of mice revealed that strain A/74/2023 induced more severe pulmonary and neurological lesions than A/75/2023. Whole-genome sequencing and phylogenetic analysis demonstrated close relatedness to HPAI H5N1 strains circulating in the Russian Far East and Japan from 2022 to 2023, with several mutations associated with mammalian adaptation, including NP-N319K and, in one isolate, PB2-E627K. According to our findings, northern fur seals (*Callorhinus ursinus*) on Tyuleniy Island acted as spillover hosts for the highly pathogenic avian influenza (HPAI) H5N1 virus of clade 2.3.4.4b. Furthermore, the high population density of fur seals and the extensive mortality observed during the outbreak highlight these animals’ potential role as another vessel for the evolution of avian influenza viruses. This study represents the first documented case of HPAI H5N1 in pinnipeds in the North Pacific region and supports previous reports indicating that pinnipeds, including northern fur seals, are highly susceptible to HPAI H5N1 clade 2.3.4.4b viruses.

## 1. Introduction

Orthomyxoviridae is a family of negative-sense RNA viruses that includes nine genera, one of which (*Alphainfluenzavirus*) includes avian influenza viruses (AIVs), which primarily affect birds but can sometimes affect mammals, including humans. Classification of a virus strain as either low-pathogenic avian influenza (LPAI) or highly pathogenic avian influenza (HPAI) is based on the severity of symptoms in domestic chickens [1]. The viral genome consists of eight protein-coding segments. Subtypes are classified using different subtypes of the surface proteins—hemagglutinin (H) and neuraminidase (N) [2]. To date, 17 hemagglutinin subtypes (H1–H16, and H19) and 9 neuraminidase subtypes (N1–N9) have been observed in a range of bird species [3,4]. Deeper classification into clades and subclades depends on the phylogenetic parameters of hemagglutinin.

While HPAI has long been a threat to animal health, commercial agriculture, and food security, its risk to public health and wildlife conservation has increased dramatically. Isolation of AIVs in marine mammals, including seals, has been reported sporadically over the past 45 years [5]. Until 2016, all reports of AIV detected and isolated in seals were attributed to LPAI. In 2016, HPAI subtype H5N8 was isolated from grey seals in the Baltic Sea [6]. Phylogenetic analysis of the HA and NA segments revealed that the isolate belonged to the clade 2.3.4.4b group.

Interspecies transmission of the highly pathogenic influenza H5N1 virus (clade 2.3.4.4b) has now been reported in 73 mammal species in Europe, North America, South America, and Asia [7]. Seals are one of the marine mammal groups that have been significantly affected by the current H5N1 HPAI outbreak. The H5N1 2.3.4.4b virus has been reported in pinnipeds in the Americas (USA, Canada, Argentina, Chile, Peru, Uruguay, and Brazil), Europe (Germany, Russia, Denmark, The Netherlands, Poland, Sweden, South Georgia, and the South Sandwich Islands (UK)), and Antarctica (Joinville Island and Possession Island). In the northern hemisphere, the H5Nx 2.3.4.4b clade has been most commonly observed in harbor seals (*Phoca vitulina*) and grey seals (*Halichoerus grypus*) [7].

Characterization of H7N7 [8], H3N8 [9], H4N5 [10], H3 [11], and H10N7 [12] subtypes detected in seals suggests an avian origin, which is thought to result from interspecies transmission between different species of mammals or secondary spread from wild water birds. Reported outbreaks of AIV in pinnipeds have mainly resulted in fatal respiratory illness. Northern fur seals appear to be particularly susceptible to AIV infections.

Like swine, seals can also act as ‘mixing vessels’ for human and avian influenza viruses, highlighting pinnipeds’ potential to generate new reassortant influenza strains that can infect mammals and even humans. Mammalian sialic acid receptor (SAα-2,6) is widely expressed in the pulmonary parenchyma and alveolar/bronchiole epithelia of seals. Avian sialic acid receptor (SAα-2,3) is generally limited to epithelial cells of the bronchioles [9,13]. A species that can become infected with multiple influenza virus subtypes should be considered a significant threat to both wildlife and public health. Analysis of HPAI H5N1 strains identified in mammals has revealed mutations indicative of adaptation to mammals [7]. These mutations include E627K in PB2, which enhances viral replication in mammalian cells while reducing its ability to replicate in avian cells [14]. Such adaptive mutations in mammals may serve as molecular markers that can facilitate the assessment of the pandemic risk posed by circulating AIVs.

As we described in [15], in July 2023, the deaths of northern fur seals (*Callorhinus ursinus*) and Steller sea lions (*Eumetopias jubatus*) were recorded in the Far East region of the Russian Federation, particularly on Tyuleniy Island (48°30′12.2″ N, 144°37′41.2″ E), which is located in the southwestern sector of the Sea of Okhotsk (north west Pacific Ocean), near Sakhalin Island (Figure A1, Appendix A). The sandy beaches of the island are home to both northern fur seals and Steller sea lions. The 2015 IUCN Red List assessment classified the Northern Fur Seal (*Callorhinus ursinus*) as Vulnerable (VU) under criteria A2b (a population reduction of at least 30% over the last 10 years or three generations). The island’s smallness forces these marine mammals to densely congregate. The island also acts as key breeding area for many species of seabirds [16], Figure A2, Appendix A.

On Tyuleniy Island, we detected the HPAI H5N1 strains belonging to clade 2.3.4.4b, isolated from northern fur seal carcasses. These strains are closely related to HPAI virus strains detected primarily in the Russian Far East and Japan in 2022–2023 [15]. Given the emergence of this virus in marine mammals near an economically developing and fishing region in the North Pacific and the public health concern associated with the spread of avian influenza among mammals, we aimed to describe the biological and genetic properties of this virus, along with related pathological and histological findings.

## 2. Materials and Methods

### 2.1. Ethical Issues and Clinical Specimens

All sampling procedures and animal study protocols were approved by the Committee on Biomedical Ethics of the Federal Research Center of Fundamental and Translational Medicine (protocol code, 2023-07; date of approval, 15 July 2023). Experiments were conducted in Biosafety Level 3 facilities. The fur seal samples for this study were collected in accordance with all required permits and approvals issued by the regional environmental authority. The fieldwork protocols were conducted in accordance with established ethical guidelines to minimize distress and harm to the animals. Individual fur seals were carefully restrained using humane methods to minimize the time taken for sample collection. We used “Seal capture and sample collection Protocol” [17].

We recorded the death of over 3500 northern fur seals on Tyuleniy Island between 15 July 2023 and 15 August 2023. The island is less than 700 m long and 40–90 m wide, with a total area of 0.053 km^2^. We obtained a visual account of the dead animals while moving along the island. Many adult animals died in the surf or water, and it is likely that the actual number of dead animals exceeded the figure we obtained. Samples were collected from the lungs, small intestines, and livers of 2 recently deceased adult fur seals. Material for virus isolation and molecular diagnostics was frozen at −20 °C and placed in liquid nitrogen for 2 days for storage and transportation to our laboratory.

We used vacuum tubes to collect blood samples from the caudal gluteal vein of 11 living fur seal pups. After separation, the serum was collected in polyethylene tubes and frozen at −20 °C. Blood serum was delivered to the laboratory in a styrofoam container with refrigerants (the temperature upon opening did not exceed +8 °C). In the laboratory, the blood serum samples were stored at −24 °C until analyzed.

### 2.2. Detection and Isolation of Viruses

Frozen tissues (approximately 60 mg) obtained during the necropsy were homogenized in 500 μL phosphate-buffered saline (PBS) using ceramic beads in a FastPrep-24 5G homogenizer (MP Biomedical, Navi Mumbai, India) and then centrifuged at 12,000× *g* RCF for 5 min. RNA was extracted from 140 μL of this clarified tissue supernatant using the QIAamp Viral RNA Mini Kit protocol (Qiagen, Hilden, Germany). The extracted RNA was then tested via reverse transcription-PCR (RT-PCR) using the commercial Amplisens-H5 kit (Interlabservice LLC, Moscow, Russia).

The AIV from the PCR-positive organs was isolated in embryonated chicken eggs, as described previously [15], and in a Madin–Darby canine kidney (MDCK) cell culture. For this purpose, cells were cultured in Eagle’s minimal essential medium (MEM) supplemented with 10% fetal bovine serum and penicillin (100 IU/mL)/streptomycin (100 μg/mL). Then, 100 μL of cleared supernatant from homogenized seal tissue was inoculated onto an 80% confluent monolayer of MDCK cultured in MEM supplemented with 0.2% BSA, 2 μg/mL of TPCK-trypsin, and penicillin (100 IU/mL)/streptomycin (100 μg/mL). Cells were observed daily to analyze the cytopathic effect. The 50% egg infectious dose (EID_50_) and the 50% tissue culture infectious dose (TCID50) for MDCK cells were determined as recommended in [18].

### 2.3. Serological Analysis

The antibodies to A/Northern_fur_seal/Russia_Tyuleniy/74/2023 and A/Northern_fur_seal/Russia_Tyuleniy/75/2023 in blood sera from 11 living fur seal pups were determined using the hemagglutination inhibition test (HAI). The sera samples were pretreated with RDE to inactivate nonspecific inhibitors of hemagglutination in accordance with the protocol recommended by the WHO [18]. The HAI was also performed according to the standard procedure recommended by the WHO [18]. For this purpose, a series of 2-fold dilutions of sera samples was prepared in 96-well plates; the ratios for the dilutions ranged from 1:10 to 1:1280. An antigen dilution containing 4 HU (hemagglutination unit) of A/74/2023 or A/75/2023 was added to each well with the serum dilution. The plates were incubated at room temperature for 30 min. Then, a 0.75% suspension of horse red blood cells was added to each well. The plates were incubated for 1 h. The highest dilution of the serum that completely inhibited hemagglutination was taken as the HI titer. The serum was considered positive at a titer ≥1:40. The blood serum of mice infected with the studied viruses was used as a positive control.

### 2.4. Pathogenicity Assessment In Vivo

The IVPI (Intravenous Pathogenicity Index) test was performed to assess the pathogenicity of the virus in chickens according to the standard procedure recommended by the World Organisation for Animal Health (WOAH) [19]. Briefly, groups of 10 specific pathogen-free (SPF) 6-week-old chickens were intravenously inoculated with 0.1 mL of a 1:10 dilution of allantoic fluid containing the viruses, and they were monitored daily for clinical signs and mortality over a 10-day period. Each bird was scored based on the severity of clinical signs according to the WOAH scoring system, and the IVPI value was calculated as the mean score per bird. According to the criteria set by the WOAH for evaluating pathogenesis in chickens, a virus with an IVPI value greater than 1.2 was defined as HPAIV.

We used laboratory mice to assess the pathogenicity of the two H5N1 clade 2.3.4.4b influenza viruses (A/Northern_fur_seal/Russia_Tyuleniy/74/2023 and A/Northern_fur_seal/Russia_Tyuleniy/75/2023) isolated with respect to mammals. For this purpose, we used male SPF mice of the Balb/c line (from the SPF Vivarium of the Institute of Cytology and Genetics, Siberian Branch, Russian Academy of Sciences) weighing 20 ± 2 g. During the transfer from the vivarium, a visual animal examination was carried out; no abnormalities were found. The animals were randomly assigned to groups. Mouse lethal dose 50% (MLD_50_) was determined as previously reported [20]. The animals were infected intranasally at a dose of 10 MLD_50_/50 μL per mouse inoculum (25 μL in each nostril). The mice were lightly pre-anesthetized with Zoletil. A group of fifteen 9-week-old BALB/c mice was used for each strain of the virus. The control group, consisting of 15 mice, was inoculated with PBS. The animals’ weights were determined every day. Mice were euthanized if they lost more than 25% of their initial body weight. Euthanasia and collection of internal organs for the determination of virus titers and histology were conducted for three animals on days 3, 6, 10, 15, and 21 d.p.i.

Samples of internal organs (lungs, brain, heart, liver, kidneys, spleen, and intestines) were taken for comparative virological analysis. The infectious activity of the virus was determined via titration of 10% mouse organ homogenates in an MDCK cell culture. Accordingly, plates with a daily cell monolayer were infected with tenfold dilutions of organ homogenates, starting with a dilution of 10^−1^. The infectious titer of the virus was expressed as lg TCID_50_/mL.

The reliability of differences in the normal and abnormal distributions was determined according to Student’s criterion and the Mann–Whitney criterion, respectively. We used Statistica 10 (StatSoft, Tulsa, OK, USA), Excel (Microsoft Office, Redmond, WA, USA), and GraphPad Prism v.10.5.0 0 (GraphPad Software, San Diego, CA, USA).

### 2.5. Histological Analysis

On days 3, 6, and 10 post-infection, the animals were withdrawn from the experiment, and their organs were collected for PCR and histological examination. Fixation and further preparation of organs for histological examination via light microscopy were performed according to the standard method [21]. The lungs and brains of the mice were assessed based on the following parameters: blood vessel status; the presence, nature, and location of inflammatory infiltrates; changes in the main organ structures; and the presence/absence of necrosis, edema, or hemorrhaging. The area of pathologically altered regions, as well as the severity of pathological changes, was assessed using a score from 1 to 4 (Table 1).

The group arithmetic mean (M) and standard error of the mean (SEM) were calculated for all the quantitative data obtained during the semiquantitative assessment of organ condition in the experimental groups. The data were statistically processed using Statistica 10 (StatSoft, USA). The reliability of difference was determined according to Student’s criterion. Differences were determined at a significance level of *p* ≤ 0.05.

### 2.6. Genome Sequencing and Phylogenetic Analysis

Complete genome NGS sequencing was performed using the Illumina MiSeq platform and associated reagent kits (also from Illumina), employed according to the manufacturer’s instructions. RNA was extracted using the QIAamp Viral RNA Mini Kit (Qiagen, Germany). Whole-genome amplification was performed using a modified version of the protocol reported by Bin Zhou. DNA libraries were prepared using a Nextera DNA Flex Library Prep kit (Illumina, San Diego, CA, USA). The DNA libraries were sequenced with a reagent kit, version 3 (600-cycle), via a MiSeq genome sequencer (Illumina). The consensus sequences were generated using the Bowtie software 1.2.3.

Nucleotide sequences were deposited in the GISAID database (EPI_ISL_18554237, EPI_ISL_19080209, and EPI_ISL_19080210). Multiple alignment was performed using MUSCLE, and editing, including translation of the nucleotide sequences into amino acid sequences, was performed using the BioEdit v. 7.2.5 and UGENE v. 50.0 software. Initial maximum-likelihood phylogenies for each of the gene segments were generated with RAxML [22] using the general time-reversible nucleotide substitution model. Final dendrograms were generated and visualized with MEGA5. Bootstrap support values were generated using 1000 rapid bootstrap replicates.

## 3. Results

### 3.1. Descriptive Epidemiology and Necropsy Findings

The carcasses of two northern fur seals found over the course of 15 July–15 August 2023 were subjected to postmortem examination. During this period, we observed the death of more than 3500 pinnipeds on the island. Since many adult animals died in the surf and open water, it is likely that the actual number of animal deaths exceeds the number counted. The first pup death was detected on 4 August; subsequently, pup mortality became widespread. During the observation of infected animals, two stages of disease progression were identified from the onset of symptoms. Stage 1 lasted 6–8 h and featured fever, lethargy, confusion, and disorientation, while Stage 2 lasted 2–4 h and featured convulsions and death. All the dying animals that we managed to observe exhibited these effects.

### 3.2. Pathological Picture During the Necropsy

The seals were in a well-fed condition. Foamy or mucous discharge from the nose was found in some carcasses of recently deceased pups. The results of the necropsy showed noticeable macroscopic lesions limited to lymphadenomegaly (submandibular and mesenteric lymph nodes), reddish foam in the tracheal lumen, and congestion in the lungs. Interstitial pneumonia was observed either in conjunction with or separately from alveolitis (Section A.3).

### 3.3. Virology Assessment

Samples were taken from the lungs, small intestines, and livers of two deceased northern fur seals. Using real-rime PCR, we detected AIV of the H5 subtype in the lungs and small intestine of the first animal and in the lungs and liver of the second animal. The AIV from the PCR-positive organs was isolated in embryonated chicken eggs and an MDCK cell culture.

The 50% egg infectious dose (EID_50_) and the 50% tissue culture infectious dose (TCID_50_) were determined for the two viruses isolated: A/Northern_fur_seal/Russia_Tyuleniy/74/2023 (hereafter A/74/2023) and A/Northern_fur_seal/Russia_Tyuleniy/75/2023 (hereafter A/75/2023). Since isolates A/74/2023 and A/74-2/2023 were obtained from different organs (the small intestine and lung, respectively) from the same animal and their genomes are identical, this study focuses on one of them (A/74/2023). Consequently, the experiments described herein were conducted using two isolates from different animals: A/74/2023 and A/75/2023. Both A/74/2023 and A/75/2023 replicated effectively in embryonated chicken eggs (10^8.5^ and 10^8.0^ EID_50_/mL, respectively) and MDCK cells (10^6.6^ and 10^6.4^ TCID_50_/mL, respectively).

Both A/74/2023 and A/75/2023 viruses were evaluated for their relative virulence in chickens and BALB/c mice. A pathogenicity analysis of these two viruses in chickens revealed that the A/74/2023 and A/75/2023 viruses killed all 10 chickens within 48 h and yielded an IVPI ranging from 2.7 to 3.0. This result indicates that all the viruses were highly pathogenic toward chickens.

Both H5N1 viruses killed the mice without prior host adaptation, with an evaluated 50% mouse lethal dose (MLD_50_) of 1.9–2.5 log_10_EID_50_/mL. H5N1 influenza viruses with an MLD_50_ ranging from 1.6 to 5.8 log_10_EID_50_/mL can be classified as highly virulent toward mice [20]. Thus, both H5N1 virus strains isolated from seals were highly pathogenic toward mice.

### 3.4. Pathogenicity of Viruses in Laboratory Mice

To assess the pathogenesis of the two H5N1 clade 2.3.4.4b influenza viruses isolated (A/74/2023 and A/75/2023) for mammals, we analyzed them using laboratory white SPF mice of the Balb/c strain. For each strain of the influenza virus, a group of 15 white Balb/c mice was formed. We planned to collect material on days 3, 6, 10, 15, and 21 after inoculation, but all the animals died within 15 days (Figure 1, Appendix B). The experiment was stopped on the 11th day after inoculation.

We assessed the dynamics of weight changes in all the mouse groups in the experiment. Mice that lost more than 25% of their initial body weight were euthanized. No live animals were found during weighing. The two dead animals exhibited 25.3% and 33.3% weight loss after death. The relative weight of the control group fluctuated around 100% of the initial weight (99.9 ± 0.8% SD, and in the range of 98.6–103.6%) during the experiment. In group A/75/2023, active weight loss began on day 5 after inoculation (97.3 ± 2278%) and reached 89.2 ± 2.41% of the initial weight of the surviving animals on day 10 after inoculation. In group A/74/2023, active weight loss began on day 3 after inoculation (93.6 ± 5.16%) and reached 76.4 ± 6.32% on day 7 (in the range of 66.7–82.2%, with 66.7% dead animals) (Figure 2, Table A2 Appendix B). Mouse weight in group A/74/2023 began to significantly differ from the control group and group A/75/2023 by 3 d.p.i. (*p* < 0.01). Mice weight in group A/75/2023 significantly differed from that of the control group starting on 6 d.p.i. (*p* < 0.01).

In order to study the pathogenicity of strains A/74/2023 and A/75/2023, 9-week-old BALB/c mice were infected intranasally with 10 MLD_50_ of the virus. On days 3, 6, and 10 post-infection, three animals from each group were euthanized, and samples of the internal organs (lungs, brain, heart, liver, kidneys, spleen, and intestines) were taken for comparative virological analysis.

In the lungs of the mice infected with 10 MLD_50_ of the A/75/2023 strain, the virus was detected only on the 6th day in a titer of 4.04 ± 0.15 log_10_ TCID_50_/mL (Figure 3). At the same time, in the lungs of the mice infected with 10 MLD_50_ of the A/74/2023 strain, the virus was detected on both the third and sixth days, in titers of 4.96 ± 0.15 log_10_ TCID_50_/mL and 4.83 ± 0.18 log_10_ TCID_50_/mL, respectively. By the 10th day, all animals infected with this strain had died, so it was impossible to determine the infectious titer in the organs. On the third day, the virus was not detected in the brain in animals from either group, while on the sixth day, the virus was detected in all the animals infected with the A/74/2023 strain in a titer of 3.63 ± 0.15 log_10_ TCID_50_/mL. At the same time, in the group of animals infected with the A/75/2023 strain, the virus was detected in only one animal out of three in a titer of 3.88 ± 0.21 log_10_ TCID_50_/mL. On the 10th day, the virus was not detected in the mice in this group. The virus was not detected in other organs in mice from either group.

### 3.5. Results of the Pathomorphological Examination of Lung and Brain Samples

#### 3.5.1. Pathological Changes in the Lungs of Mice Infected with Strains A/74/2023 and A/75/2023

We observed the dynamic development of an infectious process characteristic of a respiratory infection in the lungs of the mice from both experimental groups. In the group of animals infected with strain A/74/2023, on the third day, a reaction characteristic of the onset of the infectious process developed (Figure 4 and Figure A1). In general, the terminal part of the respiratory section of the lungs was not damaged; the alveoli were open and free, although there was thickening of the interalveolar septa in some places. The structure of bronchi of various sizes underwent pathological changes. In some areas of the bronchi, we observed large accumulations of cellular debris, which had formed as a result of the destruction of the epithelial lining of the mucous membrane and underlying tissues. In the interstitium, we observed foci of leukocyte infiltrate and hemorrhaging. By the sixth day, we observed a rapid spread of the inflammatory process to the respiratory section, accompanied by a larger area of lung tissue damage (Figure 4 and Figure A2). Multiple foci of necrosis were observed in bronchial walls. Large atelectases and pronounced edema in the interstitium appeared, and interstitial pneumonia developed.

When the mice were infected with strain A/75/2023, the infectious process proceeded in a milder form compared with the infection caused by A/74/2023 throughout the entire observation period (Figure 5, Table A3, Appendix B). By the third day, signs of an inflammatory reaction appeared (Figure 4, Table A1). Thickening of the interalveolar septa, desquamation of epithelial cells in the bronchial tree, and moderate leukocyte infiltrate, mainly perivascular, were observed. By the sixth day, mucus and cellular debris accumulated in the bronchi of various sizes as a result of intensive desquamation of the epithelium (Figure 4, Table A2). Large accumulations of leukocyte infiltrate and atelectasis were found in the interstitium in certain places. At the same time, in some areas of the lung, pathological changes were less pronounced and characterized only by filling of the blood vessels with blood and thickened septa in the alveoli. By the 10th day, the picture visually improved (Figure 4, Table A3). Even though there were still areas exhibiting pronounced signs of inflammation (atelectasis, perivascular infiltrate, and cellular detritus in the bronchi and bronchioles) in some places, a significant part of the structure of the lungs was restored. In these areas, the alveoli were open, and the respiratory section and airways were free.

Thus, in the animals infected with the A/75/2023 strain, inflammation occurred gradually, with vivid manifestations already occurring by the third day and the peak of infection occurring on the sixth day. By the 10th day, the processes of restoration of organ structures were observed. When the mice were infected with the A/74/2023 virus, significant pathomorphological changes in both the respiratory section and the airway were observable by the sixth day. These changes led to a sharp deterioration in the condition of the lungs of the mice in this group.

#### 3.5.2. Pathological Changes in the Brain in Mice Infected with Strains A/74/2023 and A/75/2023

In both groups of mice, pathological changes in the brain structures were observed during all the observation periods, primarily in the cerebral hemispheres and cerebellum. The nature of the damage to the brain structures was similar, but when the mice were infected with the A/74/2023 strain, pathomorphological changes progressed more intensively as the infection progressed. Thus, in the mice from this group, on the third day, we observed perivascular edema, erythrocyte sludge in the blood vessels, and massive leukocyte infiltrate in the cerebral cortex, and large areas of dead neurons were also found. In the ganglion layer of the cerebellum, Purkinje cells were in a state of chromatolysis (Figure 6(A1)). By the sixth day, we observed destruction of the ganglion and granular layers of the cerebellum, Purkinje cells, hemorrhages, and necrosis. In the cerebral hemispheres, there were large areas of destroyed neurons, along with edema of the pia mater (Figure 6(A2)).

In the group of mice infected with the A/75/2023 virus, the picture observed in the cerebral hemispheres on the third day was similar to that for the other group, while no significant changes were found in the cerebellum (Figure 6(B1)). By the sixth day, lesions spread to the structures of the cerebellum: edema appeared in the granular and ganglion layers, and necrosis was detected in the molecular layer, with Purkinje cells in a state of chromatolysis (Figure 6(B2)). On the 10th day, the picture generally remained unchanged: destructive changes were observed in the cerebellum, with necrosis in the ganglion and granular layers. In the cerebral cortex, multifocal necrosis, as well as perivascular edema, was observed (Figure 6(B3)).

Thus, progressive destructive changes in the brain in both groups of mice were observed throughout the observation period, and although the pathomorphological aspect of the cerebral hemispheres did not differ in dynamics, differences were clearly visible in the cerebellum. In the mice infected with the A/74/2023 virus, changes in the cerebellar structures were already evident on the third day, and partial destruction of the cerebellar layers was observable by the sixth day. Our assessment of brain structure lesions showed that the pathological changes in tissue were more expressed 6 dpi in the group of mice infected with A/74/2023 (Figure 7; Table A4, Appendix B).

### 3.6. Analysis of the Presence of Antibodies to Influenza Virus Subtype H5

We tested 11 blood sera collected from living fur seal pups on Tyuleniy Island during the outbreak of the virus for the antibodies specific to A/74/2023 and A/75/2023 using the hemagglutination inhibition test. Our findings indicated that all the sera samples were negative for both A/74/2023 and A/75/2023 suggesting the animals were not exposed to the virus.

### 3.7. Sequencing and Phylogenetic Analysis

The initial phylogenetic analysis was presented in our previous brief publication on HPAI in fur seals [15]. However, since then, new sequences have been published in the GISAID database, including not only virus genomes that circulated later, in 2024–2025, but also those from 2023. Thus, we decided to update the phylogenetic analysis, taking the new data into account (Figure 8).

For the updated phylogenetic analysis, we used the nucleotide sequences of the HPAI H5N1 genome segments of viruses found in pinnipeds presented in the GISAID database as well as the sequences found as a result of a BLAST v.2.17.0 analysis (100 closest) for northern fur seal viruses. Based on the current data, we found that the viruses of northern fur seals form a common phylogenetic group with HPAI H5N1, whose sequences were not present in GISAID when the 2024 publication was being prepared. These viruses were detected in Primorye in autumn 2022 and summer 2023 as well as on Sakhalin Island in summer 2023, at the same time as the outbreak of HPAI H5N1 among northern fur seals.

Thus, the HPAI H5N1 isolated from northern fur seals belongs to a viral population that probably entered the Russian Far East from Japan during spring bird migration and subsequently circulated in Primorye and on Sakhalin Island for several months. According to phylogenetic analysis, viruses similar to those isolated from northern fur seals were subsequently detected in Hokkaido and Sakhalin (including in poultry) in January–February 2024.

The analysis of amino acid substitutions was also expanded. For isolates of HPAI viruses obtained from northern fur seals (*Callorhinus ursinus*) on Tyuleniy Island, we conducted a search for amino acid substitutions previously reported to be associated with mammalian adaptation and increased pathogenicity (Table 2).

In the PB2 protein, we screened for the following mutations: T271A [23], E627K [24,25], E627V [26], D701N [27], and the combination L89V+G309D+T339K+R477G+I495V+A676T [28]. Among the Tyuleniy Island viruses, only T271A and D701N were not detected in any isolate. A single isolate contained the E627K substitution.

In the PB1 protein, substitutions D3V [29] and D622G [30] were identified. In the PA protein, substitutions S37A [31] and N383D were present, while S224P [32] was absent. In the NP protein, we detected the substitution N319K [33]. Mutations P42S, L103F, I106M, K55E+K66E+C138F, and V149A were identified in NS1 [34,35,36,37]. Additionally, substitutions N30D, I43M, and T215A were identified in M1 [38,39].

We further analyzed the prevalence of amino acid substitutions associated with mammalian adaptation and pathogenicity in H5Nx viruses circulating in Russia, Japan, and Korea (the Pacific region) from 2020 to 2025.

Analysis of PB2 revealed that among 2382 amino acid sequences, the substitution T271A/I was present in only 8 cases. The E627K substitution was found in 38 sequences, of which only 8 originated from mammalian viruses. We obtained two of these isolates from different organs of the same northern fur seal. The E627V substitution was not detected. The D701N substitution was identified in ten sequences, five of which originated from mammalian viruses. At PB2 position 89, alternative substitutions to L89V (namely, L89I and L89M) were identified in 10 cases. Substitutions G309D, T339K, and R477G were present in more than 99.5% of the sequences. While I495V and A676T were also dominant, their frequencies were somewhat lower than those of the other prevailing mutations: 99.1% and 93.9%, respectively.

In PB1, the substitutions D3V and D622G were found in 100% and 99.9% of the 2376 sequences, respectively.

In PA, the substitutions S37A and N383D were present in 100% and 99.34% of the 2380 sequences, respectively, while S224P was observed in 84.5%.

The N319K substitution was detected in 561 of the 2360 NP sequences (23.8%).

In NS1, substitutions at positions 42, 103, 106, 55, 66, 138, and 149—associated with mammalian adaptation and increased pathogenicity—were dominant in 84.6% to 100% of cases. The three substitutions in M1 (N30D, I43M, and T215A) were also present in the vast majority (99.9–100%) of HPAI virus sequences circulating in Russia, Korea, and Japan between 2020 and 2025.

In addition to identifying biologically significant mutations, we conducted a comparative analysis of all three complete HPAI genome sequences from Seal Island. The sequences A/74/2023 and A/74-2/2023, isolated from different organs from the same animal, are predictably identical, whereas the genome segment sequences of isolate A/75/2023 exhibit several amino acid substitutions. In addition to identifying biologically significant mutations, we conducted a comparative amino acid analysis of the complete HPAI genome sequences obtained from Tyuleniy Island. Six substitutions were identified in A/74/2023 relative to A/75/2023: PB2-E627K, PA-N222D, HA-V328I, HA-L505P, NP-L419P, and NS1-V155A.

## 4. Discussion

All continents (except Australia) and most countries have experienced unprecedented mortality in wild and domestic birds due to the highly pathogenic avian influenza (HPAI) A(H5N1) clade 2.3.4.4b [40,41]. Infections affecting tens of thousands of wild birds representing at least 112 species, including many seabirds, have been reported [40,42,43].

Since 2020, the highly pathogenic avian influenza virus H5N1 clade 2.3.4.4b (HPAIV) has had a significant impact not only on wild birds but also on mammals worldwide, with confirmed detection in millions of animals and over 500 species [7].

Until 2014, all reports of AIV detection in seals corresponded to low-pathogenicity viruses [44,45,46]. Nevertheless, most reported AIV outbreaks caused fatal respiratory diseases, with harbor seals being found to be particularly susceptible to infection. The highly pathogenic AIV clade H5 2.3.4.4b (HPAIV) was first detected in seals in 2016. Many recent pinniped mortality events have been attributed to HPAIV 2.3.4.4b spread by wild bird species.

The 2.3.4.4b H5N1 virus has been reported in pinnipeds in North and South America (USA, Canada, Argentina, Chile, Uruguay, and Brazil) and Europe (Germany, Russia, Denmark, South Georgia, and the South Sandwich Islands (UK)) [5,46]. We found a highly pathogenic influenza A(H5N1) virus of clade 2.3.4.4b associated with seal mortality events on Tyuleniy Island [15].

The pathological aspects of the macroscopic lesions of the internal organs of dead northern fur seals were identical: foamy discharge from the nose and trachea, lymphadenomegaly, heterogeneous and congested lung surfaces, and interstitial pneumonia. Similar macropathological changes have been described in many cases and can now be considered “standard” for damage inflicted by the H5N1 virus clade 2.3.4.4b on northern fur seals [47,48].

Viruses were isolated from the lungs and small intestines of two dead animals. We previously showed that viruses isolated from different organs of one animal were genetically identical. To study the properties of the isolated viruses, we selected one virus from two different animals: A/75/2023, isolated from the lung of a northern fur seal, and A/74/2023, isolated from the small intestine of another member of the species. Pathogenicity analysis of these two viruses in chickens revealed that the A/74/2023 and A/75/2023 viruses killed all 10 chickens within 48 h and yielded an IVPI ranging from 2.7 to 3.0. This result indicates that all viruses were highly pathogenic toward chickens.

We studied the biological properties of the isolated viruses by evaluating the pathogenesis of the selected strains using the laboratory white SPF mouse model of the Balb/c line. Both H5N1 viruses killed the mice without prior host adaptation, with an evaluated 50% mouse lethal dose (MLD_50_) of 1.9–2.5 log_10_EID_50_/mL. H5N1 influenza viruses with an MLD50 ranging from 1.6 to 5.8 log_10_EID_50_/mL can be classified as highly virulent toward mice. Regardless of MLD_50_ determination, the experiments conducted on the pathogenesis of influenza virus strains A/74/2023 and A/75/2023 showed that the mice infected with the A/74/2023 virus experienced mass mortality on days 7–8 after infection, while the mice infected with the A/75/2023 virus experienced gradual, more prolonged death on days 7 to 10. The death of the mice did not allow a long-term assessment of pathogenicity, and the experiment had to be halted on day 11 after infection.

In the lungs of the mice from both experimental groups, dynamic development of the infectious process characteristic of a respiratory infection was observed. However, when the mice were infected with the A/75/2023 strain, the infectious process proceeded in a milder form throughout the entire observation period (10 days). The observed lack of virus replication in the lungs on day 3 for A/75/2023, followed by detection on Day 6, and varied virus titers in other tissues of the infected mice, can be explained by several factors related to the virus’s replication dynamics and host response. Early in infection, virus replication may be below the detection threshold or delayed due to initial immune responses or viral entry kinetics. By Day 6, replication increases as the virus successfully establishes an infection and spreads in the lung tissue. Differential replication kinetics in various tissues may reflect tissue-specific viral tropism, immune clearance, or sampling variability.

When infected with the A/74/2023 virus, by the sixth day, significant pathomorphological changes occurred in both the respiratory tract and airways, leading to a sharp deterioration in the condition of the lungs of the mice in this group. In both groups of mice, pathological changes in the structures of the brain were observed in all observation periods. The nature of the damage to the brain structures was similar, but when infected with the A/74/2023 strain, pathomorphological changes progressed more intensively during the course of the infection. We could not assess the significant differences between the strains studied. These are preliminary data, and they should be clarified in further studies using large samples of experimental animals, various doses of infection, and reverse genetics methods. Strains from clade 2.3.4.4b have been reported to cause severe systemic infections in both wild hosts [49] and laboratory animals after forced injection [50,51,52]. Phylogenetic analysis has clearly demonstrated that the highly pathogenic avian influenza virus (HPAI) H5N1, belonging to clade 2.3.4.4b, continues to actively evolve and expand its host range, affecting not only birds but also various mammalian species.

It can be concluded that HA influenza viruses isolated from mammals, particularly pinnipeds, do not form a single clade but are instead distributed throughout the phylogenetic tree. At the same time, they cluster with HA viruses isolated from birds. This information suggests that mammals have been infected repeatedly and independently in different geographical regions and at different times. Within these clusters, however, HA influenza viruses from mammals—specifically from seals on Tyuleniy Island and Sakhalin—form monophyletic groups of sequences. These sequences, as indicated by their short branch lengths, exhibit minimal genetic divergence, suggesting subsequent local transmission of the virus within dense populations of marine mammals. We believe that the mass mortality of animals and the detection of the virus carrying the PB2-E627K substitution (one of the markers of mammalian influenza viruses) may be a consequence of virus circulation among fur seals, primarily through local mammal-to-mammal transmission, rather than through more than 3500 direct transmissions from birds. Regarding another outbreak, colleagues from Argentina also concluded that the virus was transmitted within pinniped populations [47].

It is noteworthy that viruses from North America—specifically those found in bald eagles, northern pintails, and gulls—belong to the same phylogenetic group as viruses from Russia and Japan. This phylogenetic evidence supports the hypothesis that the virus was introduced to North America via Alaska along the Pacific migratory route used by birds traveling from East Asia and then transmitted locally [53].

According to our analysis of sequences from the GISAID database, amino acid substitutions in H5Nx avian influenza viruses—previously reported in the literature as being associated with mammalian adaptation and increased pathogenicity—are widely distributed in the current population of highly pathogenic H5Nx viruses of clade 2.3.4.4b, regardless of host species. This widespread presence may, at least in part, explain the sharp increase in influenza outbreaks among mammals observed in recent years.

Of all the substitutions considered (Table 1), only three were not dominant in HPAI H5Nx viruses detected in birds, mammals, and environmental samples from Russia, Korea, and Japan between 2020 and 2025. Specifically, the T271A/I substitution in PB2 was absent in HPAI viruses from northern fur seals on Tyuleniy Island; N319K in NP was identified in two isolates; and E627K in PB2 was found in two of three isolates obtained from different organs from the same animal.

Notably, in 2023 and 2025, researchers also detected HPAI viruses in fur seals from Sakhalin (Russia) and harbor seals from Hokkaido (Japan) [54]. Phylogenetic analysis revealed that these viruses were closely related to those we identified on Tyuleniy Island. Thus, concurrently with circulation on Tyuleniy Island, the virus caused an outbreak in Sakhalin, followed by another outbreak in Hokkaido 20 months later. This finding indirectly supports our hypothesis that there is localized circulation of HPAI viruses in the Pacific region, including the Russian Far East, Korea, and Japan. Importantly, aside from the detection of the PB2-E627K substitution in the virus from Tyuleniy Island, this mutation has not been observed in pinnipeds elsewhere in the Pacific region.

We further analyzed the global prevalence of the PB2-E627K mutation in mammalian viruses. E627K was identified in only 328 (6.4%) of 5144 PB2 sequences from HPAI viruses isolated from mammals worldwide between 2001 and 2025. Extending the analysis beyond the Pacific region revealed that the earliest records of E627K in mammalian HPAI viruses date back to 2004, when it was detected in cats, dogs, tigers, and leopards in Thailand (corresponding to nine sequences). Subsequently, by 2025, E627K continued to be reported intermittently in mammalian isolates, although not on an annual basis. Among HPAI viruses isolated from pinnipeds, PB2-E627K was present in 29 of 141 sequences. Therefore, this substitution remains relatively rare in mammalian HPAI viruses, particularly in those infecting pinnipeds. These findings suggest that the virus can transmit itself efficiently and cause severe disease in marine mammals while retaining the “avian” PB2-627E variant. Adaptation to novel mammalian hosts may thus occur through mechanisms other than the “classical” E627K substitution—for instance, through the combined constellation of mutations L89V, G309D, T339K, R477G, I495V, and A676T in PB2 [30]—even though E627K is a well-recognized marker of mammalian adaptation in influenza viruses.

Furthermore, based on the dates of sample collection and isolate names, it is likely that both viruses with and without the E627K mutation co-circulated during a single outbreak. For example, GISAID contains sequences of three viruses from seals in Germany collected between 15 August 2021 and 18 August 2021. Of these three PB2 sequences, two carried the E627K mutation. Additional examples include viruses from Canada, the United States, and the Crozet Islands. Given the rarity of the E627K substitution, it is plausible that it emerges de novo within infected mammalian hosts. However, influenza variants carrying E627K do not appear to gain a strong selective advantage and continue to co-circulate alongside viruses retaining the PB2-627E variant.

Prevalence analysis also showed that NP-N319K is a minor substitution, detected in only 56 (1%) of 5553 NP sequences from mammalian HPAI viruses worldwide between 2001 and 2025. The earliest reports of NP-N319K in mammalian isolates date back to 2014–2015 in regard to swine from Mexico (two sequences) and 2021 in regard to a red fox from Estonia (one sequence). Among the 56 NP-N319K mammalian HPAI sequences associated with mammalian mortality, 16 were from viruses in 2022, 22 were from 2023, 5 were from 2024, and 10 were from 2025. These findings suggest that although NP-N319K may biologically contribute to mammalian adaptation and pathogenicity, it is unlikely to play a decisive role in this process.

According to the updated phylogenetic analysis (Figure 5, Figure A4, Figure A5, Figure A6, Figure A7, Figure A8, Figure A9, Figure A10 and Figure A11 Appendix C), the sequences of all the genome segments of strains isolated from fur seals on Tyuleniy Island are closely related to virus segments detected in August 2023 in the Far East of the Russian Federation (Sakhalin and Primorye); i.e., during the outbreak on Tyuleniy Island. It is noteworthy that viruses with the PB2-E627K substitution on the phylogenetic tree are distant from the A/75/2023 virus and, more importantly, A/74/2023, which also contains this substitution. None of the closely related A/74/2023 and A/75/2023 viruses from Sakhalin and Primorye (isolated from both birds and pinnipeds) contain this substitution. Thus, we believe that the mutation has probably already occurred in the seal population. This hypothesis is also supported by the rarity of this substitution in avian influenza viruses. According to our analysis of the sequences of PB2 H5Nx viruses isolated from birds in Eurasia (based on the GISAID database), out of 18,026 sequences, only 498 (2.2%) contain the E627K substitution. At the same time, a similar analysis regarding mammalian H5Nx viruses showed the presence of this substitution in 154 (34.5%) out of 446 cases. The observations and conclusions that we were able to draw from the phylogenetic analysis of individual genome segments of isolates A/74/2023 and A/75/2023 virus are also confirmed by the analysis of genome-wide concatenated sequences (Figure A4, Figure A5, Figure A6, Figure A7, Figure A8, Figure A9, Figure A10 and Figure A11 Appendix C). They also form the Russian Far East cluster, which lacks the genome sequences of viruses carrying the PB2-E627K mutation. At the same time, the A/74/2023 and A/75/2023 viruses, unlike viruses from Sakhalin and Primorye, do not form separate monophyletic groups within the “Far Eastern” cluster. In general, based on our phylogenetic analysis, the viruses A/74/2023 and A/75/2023 are more genetically distinct than the viruses found in seals on Sakhalin by another scientific team. Unfortunately, our phylogenetic study was limited by the number of samples analyzed and their collection time: we only obtained two isolates from fur seals collected on one day, and we did not acquire any samples from birds. Thus, it is difficult to answer a number of important questions: Are the viruses detected the result of one or two spillovers? Was there seal-to-seal virus transmission? How has the virus changed in the fur seal population? Importantly, we can assume that the viruses we have identified are the result of two spillovers because although the viruses are phylogenetically similar, they do not form an unambiguously monophyletic cluster, such as viruses from birds from Sakhalin as well as from fur seals from Sakhalin. Given the mass death of fur seals, which is difficult to explain only by infection from birds, it is likely that there was mammal-to-mammal transmission. However, we cannot unequivocally state this due to the lack of data. But this hypothesis, in our opinion, is consistent with the hypothesis of the acquisition of rare for bird viruses [55], but associated with mammalian host adaptation PB2-E627K mutation after spillover from birds. There are no data on influenza viruses circulating among birds on Tyuleniy Island.

Thus, the viral dynamics in the natural reservoir (wild birds) are unclear, complicating the assessment of the spillover from birds to fur seals. However, we have the opportunity to evaluate the phylogenetic relationships of viruses from Tyuleniy Island against viruses isolated from fur seals and birds on Sakhalin Island. According to the phylogenetic analysis of both individual genome segments and concatenated sequences, HPAI viruses infecting fur seals from Sakhalin Island are phylogenetically more related to fur seal viruses from Seal Island than to avian viruses from Sakhalin Island. Thus, it can be assumed that the populations of fur seals from Tyuleniy Island and Sakhalin Island are related, the virus was transmitted by birds from one population to another, or, after independent spillovers from birds on Tyuleniy Island and Sakhalin, the viruses changed in a similar way.

In general, by summarizing the literature data, the results we obtained, and the limitations of our study, we can formulate arguments both for and against the hypothesis of seal-to-seal transmission and the evolution of the virus in the population of northern fur seals. The literature describes many cases of influenza virus (both low- and highly pathogenic) in pinnipeds [48,56,57,58].

In conclusion, HPAI H5Nx viruses can actively adapt to mammals, particularly pinnipeds—a process likely facilitated by the close ecological interactions between pinnipeds and wild birds—as evidenced by multiple outbreaks worldwide. Different mutations, for example, such as PB2-E627K and NP-N319K have emerged and occasionally become fixed, demonstrating that the adaptation process is ongoing and variable. A persistent risk of further adaptation remains. The circulation of HPAI viruses within pinniped populations, especially where multiple genetic variants (e.g., PB2-627E and PB2-627K) coexist, creates a “melting pot” for the emergence of novel, more mammal-adapted strains. Simultaneously, the close contact between wild bird populations and pinnipeds poses a significant risk for the broader dissemination of newly emerged viral variants.

Close contact between pinnipeds and wild birds, the rapid evolution of the H5N1 influenza virus, and the potential for efficient mammal-to-mammal transmission are of increasing concern due to the potential for development of a marine mammal reservoir and the public health risks associated with the pandemic potential of the virus.

Influenza may be more endemic among marine mammals than previously thought, although live viruses are difficult to extract. Avian-to-marine-mammal transmission events do not appear to be particularly rare. Such events have been documented for at least ten different subtypes in multiple regions around the globe [59]. The high population density of seals and the extensive mortality observed during the outbreaks of HPAI highlight these animals’ potential role as another vessel for the evolution of avian influenza viruses. In addition, because of limitations regarding comprehensive surveillance of pinniped haul-out areas in the Western and North Pacific, many outbreaks causing mortality are likely to be unreported. Unlike outbreaks among land mammal species and in areas with large animal and human populations, HPAI outbreaks in marine mammal populations are rarely detected in enough time to enforce disease control measures. In this region, as elsewhere, colonial breeding seabirds and colonially associated marine pinnipeds may be subjected to persistent circulation of both LPAI and HPAI viruses between species. This situation may lead to reassortment of segments of different subtypes, virus adaptation, and further dissemination of altered virus types over vast distances via migration, introducing new animal species into AIV virus circulation.

This study of HPAI A/H5N1 in Northwestern Pacific pinnipeds represents a significant contribution to our understanding of a highly pathogenic virus with serious pandemic potential for humans.

## 5. Study Limitations

Our use of high-dose intranasal infection in mice is the most critical issue and the main limitation for interpreting the results obtained. Although it is a standard experimental method used to ensure reliable infection, its artificial nature is a key limitation when generalizing findings to natural disease in wild animals such as seals. In mouse experiments regarding avian influenza H5N1, researchers often use a high dose (ranging from 1 to 10 MLD_50_) administered intranasally to ensure consistent infection and disease development in the model. This approach helps overcome the natural host resistance and variability in susceptibility among mice, resulting in reproducible data on viral pathogenicity and immune response. The high dose acts as a standard, allowing researchers to reliably evaluate the lethality and progression of infection under controlled experimental conditions.

The standard dose varies by study [19,34,35,36,37,38], but doses in the range of 1–10 MLD_50_ are commonly reported to ensure lethality and measurable disease endpoints in a mouse model. This dose is typically much higher than the natural infectious dose animals might encounter, reflecting the artificial nature of the model and the main limitation.

When extrapolating findings from this high-dose mouse model to natural transmission dynamics and disease progression in wild seals (or other wild animals), a critical limitation must be acknowledged: the artificial high-dose inoculation does not mimic the lower-dose, natural exposure route or co-factors influencing infection in wild populations. This fact limits direct extrapolation of virulence, transmission efficiency, and clinical outcomes. Results from a mouse model primarily provide mechanistic insights into viral replication, pathogenicity, and host responses under forced infection, but the ecological and epidemiological context in regard to seals must be interpreted cautiously.

The most severe limitation is the sample size of two viral isolates from a mass mortality of 3500 animals and absence of avian influenza virus sequences from birds on Tyuleniy Island. It does not allow to claim clearly the outbreak dynamic and reconstruct the spillover event(s) or rule out a common avian source for both of seal viruses. A high density of susceptible hosts and a high force of infection from a large, unsampled wild bird outbreak equally could explain the mass mortality.

## 6. Conclusions

This study of HPAI A/H5N1 in Northwestern Pacific pinnipeds represents a significant contribution to our understanding of a highly pathogenic virus with serious pandemic potential for humans. We believe that identification of mammalian-adaptive mutations, including NP-N319K and PB2-E627K, may highlight the ongoing evolution of this virus in marine mammal populations. The pathogenicity properties and lethality in mice as well as the organ tropism of the two isolated strains underscore the importance of continuous monitoring and characterization of circulating viruses. Enhanced surveillance at the wildlife–livestock–human interface in the North Pacific region is urgently needed to prevent potential spillover events and mitigate public health risks.

## Figures and Tables

**Figure 1 viruses-18-00051-f001:**
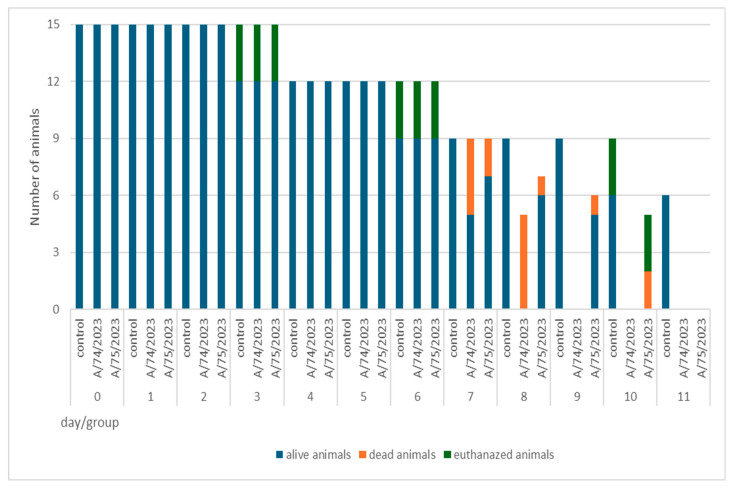
Number of animals by day of the experiment.

**Figure 2 viruses-18-00051-f002:**
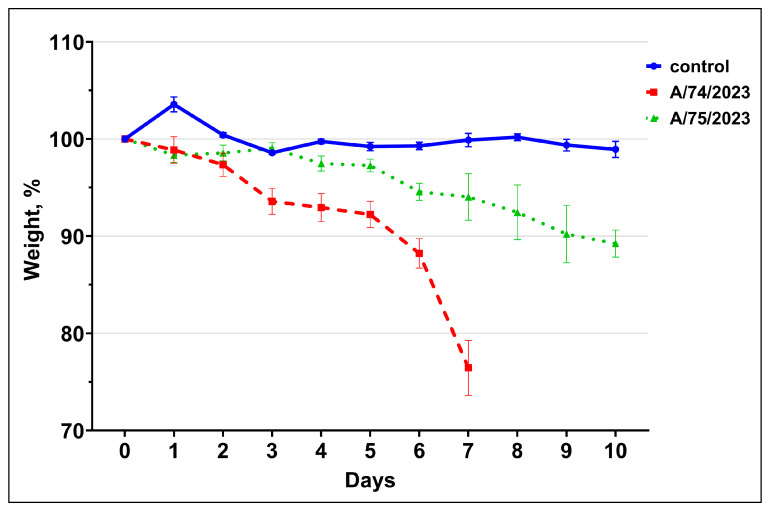
Weight changes in % (±SD) relative to the beginning of the experiment.

**Figure 3 viruses-18-00051-f003:**
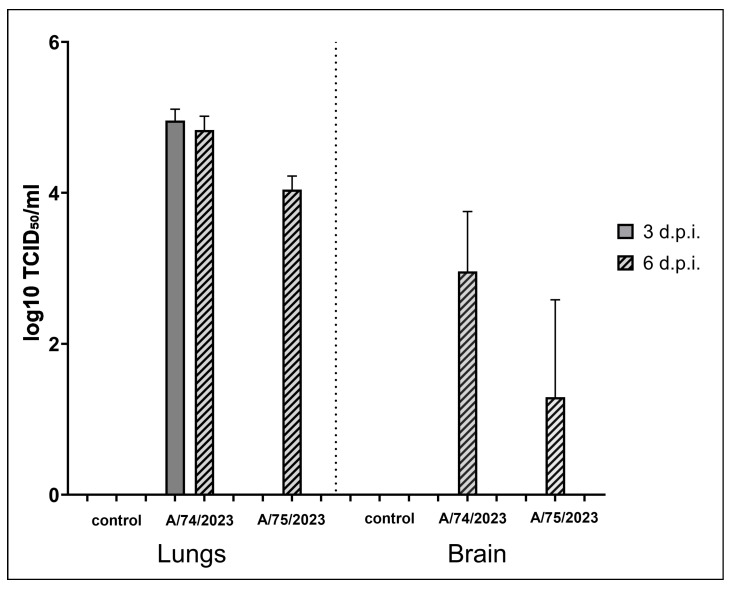
Virus detected in organs, log_10_ TCID_50_/mL (±SEM). Virus was detected in only one animal out of three (brain, A/75/2023 strain).

**Figure 4 viruses-18-00051-f004:**
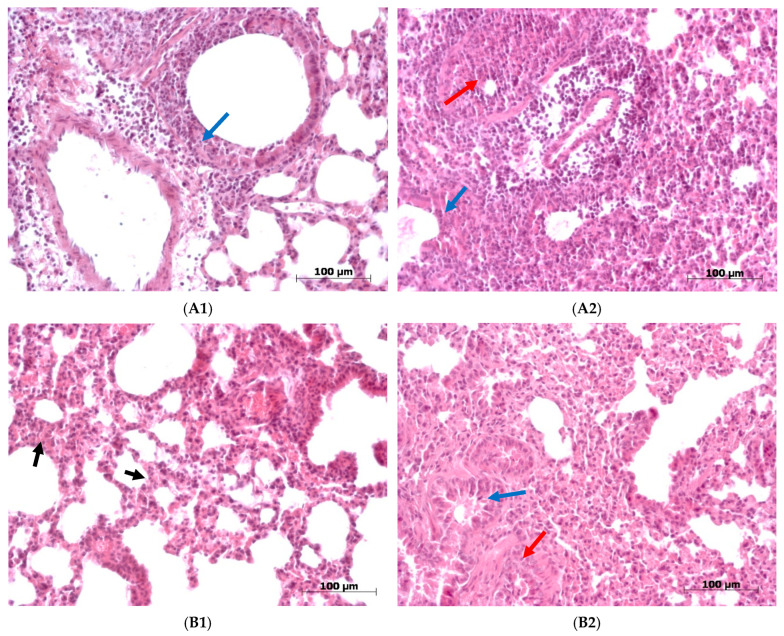
Pathological changes in the lungs of mice infected with A/74/2023 and A/75/2023 viruses in regard to dynamics. Hematoxylin-and-eosin staining was employed. Magnification: ×200. (**A**) A/74/2023. (**A1**) At 3 dpi, massive leucocyte infiltration in interstitial tissue in lungs and epithelial desquamation in bronchioles (
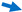
). (**A2**) At 6 dpi, in the lungs, mucus and cellular debris accumulated in the bronchi (
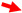
), extensive atelectasis, and pronounced edema in the interstitium and perivascular space (
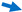
). (**B**) A/75/2023. (**B1**) At 3 dpi, in the lungs, the thickening of the interalveolar septa (
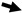
), desquamation of epithelial cells in the bronchial tree and moderate leukocyte infiltrate. (**B2**) At 6 dpi, mucus and cellular debris accumulated in the bronchi of various calibers (
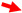
), and intensive desquamation of the epithelium (
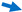
). (**B3**) At 10 dpi, thickening of the interalveolar septa of lungs (
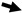
), and desquamation of epithelial cells in the bronchial tree (
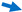
).

**Figure 5 viruses-18-00051-f005:**
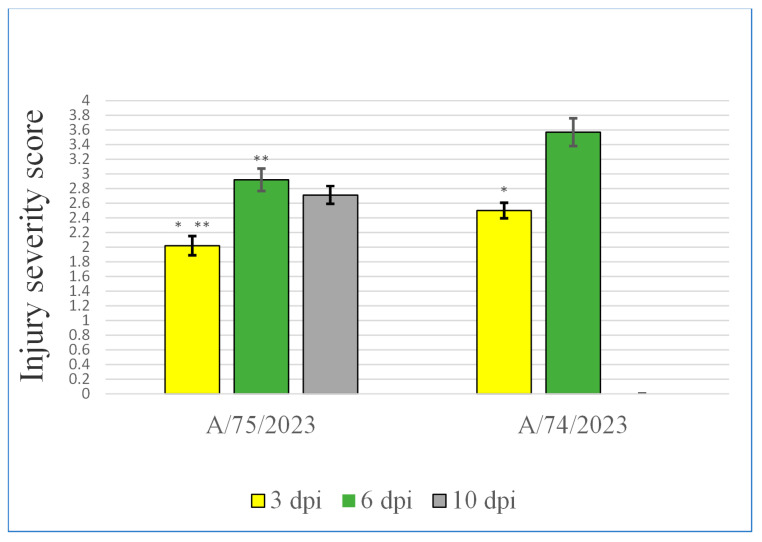
Influence of the HP AIV infection on the severity of lesions in mouse lungs (points, 1–4). Note: *—Statistically significant differences inside one group from 3 to 6 and 10 dpi (*p* ≤ 0.05). **—Statistically significant differences on the same dpi between different groups (*p* ≤ 0.05).

**Figure 6 viruses-18-00051-f006:**
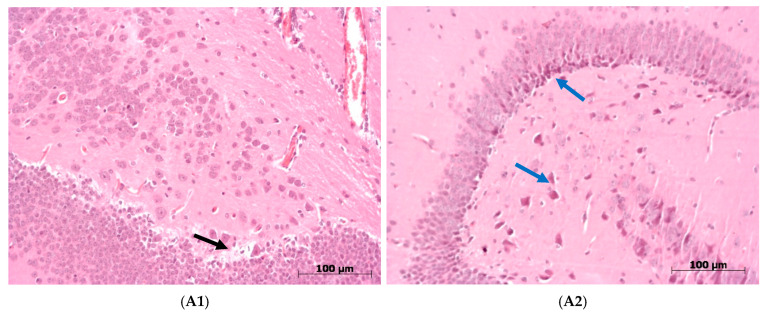
Pathological changes in dynamics in the brain in mice infected with A/74/2023 and A/75/2023 viruses (hematoxylin-and-eosin staining). Magnification: ×200. (**A**) A/74/2023. (**A1**) At 3 dpi, edema in the cerebellum (
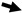
), along with erythrocyte sludge in the blood vessels. (**A2**) At 6 dpi, in brain structures, neuronal death was observed (
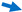
). (**B**) A/75/2023. (**B1**) At 3 dpi, erythrocyte sludge in blood vessels of brain structures and perivascular edema (
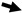
). (**B2**) At 6 dpi, edema in the granular and ganglion layers (
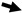
), and foci of necrosis in the molecular layer of cerebellum (
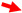
). (**B3**) At 10 dpi, multifocal necrosis (
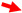
), along with perivascular edema (
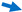
) and neuronal death (
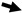
).

**Figure 7 viruses-18-00051-f007:**
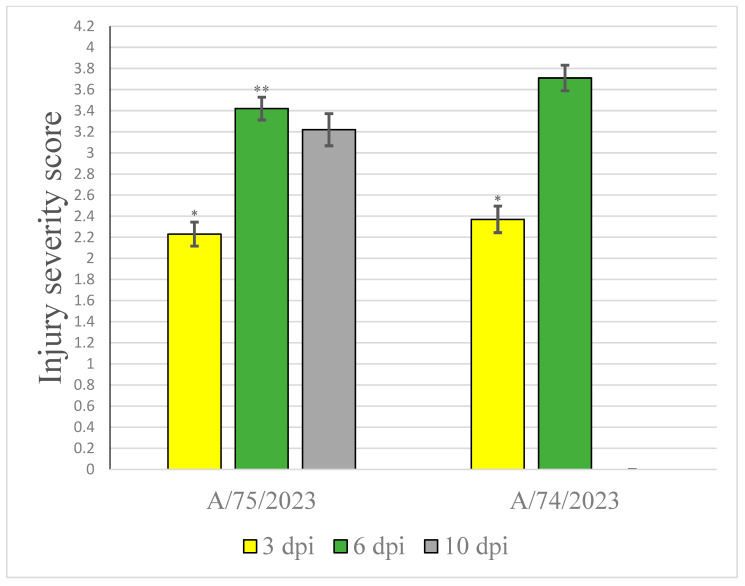
Influence of the HP AIV infection on the severity of lesions in mouse brains (points, 1–4). Note: * The statistically significant differences in one group between 3, 6, and 10 dpi (*p* ≤ 0.05). ** The statistically significant differences on the same dpi between different groups (*p* ≤ 0.05).

**Figure 8 viruses-18-00051-f008:**
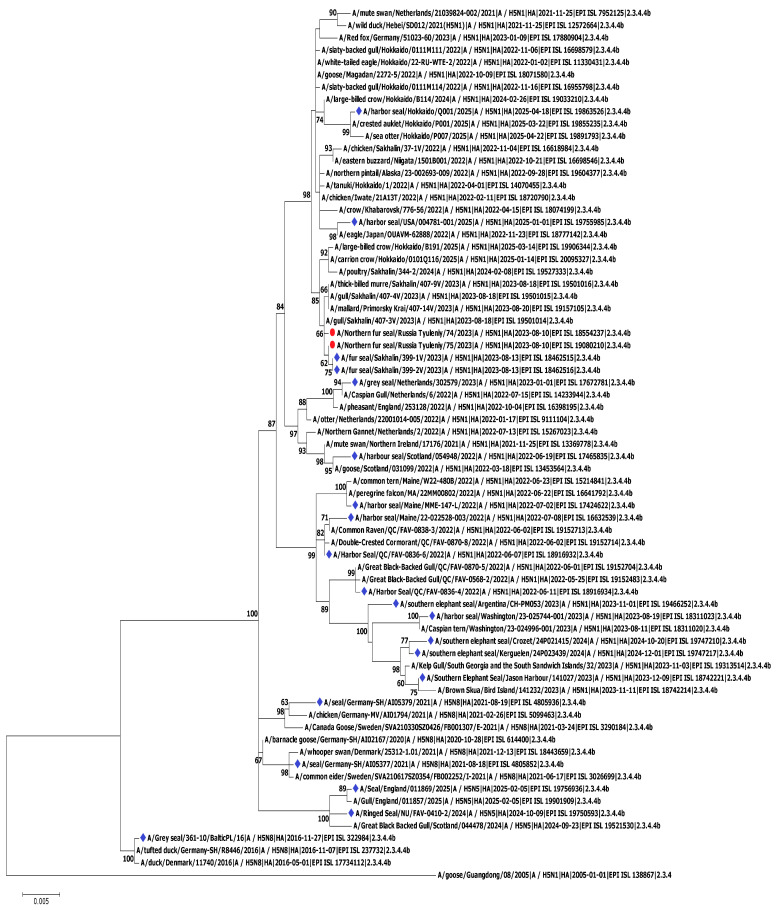
Maximum-likelihood phylogenetic tree of the hemagglutinin segment of HPAI A/H5N1 from northern fur seals on Tyuleniy Island (red dots) and other pinnipeds (blue diamonds).

**Table 1 viruses-18-00051-t001:** The assessment of significant organ damage.

Mark	Injury Severity Score	Description of Significant Assessable Damage in the Organs	
		Lungs	Brain
1	Normal	The structures of the organs in the infected mice do not differ from those in the control group.	
2	Mild	Signs of mild interstitial pneumonia, thickening of the interalveolar septa, narrowing of the alveolar lumens, filling of the blood vessels with blood, and isolated perivascular edema.	Hemodynamic disturbances, perivascular edema, and erythrocyte sludge in the cerebral vessels.
3	Moderate	Severe desquamation of the epithelium in the bronchial tree, accumulation of detritus and mucus in the lumen of the bronchi, perivascular and peribronchial edema, and leukocyte infiltrate in the interstitium and perivascularly.	Death of neurons, edema in brain structures, leukocyte infiltrate in the interstitium and perivascular space, and pinpoint hemorrhages.
4	Severe	Severe interstitial pneumonia, multiple interstitial edemas, extensive areas of atelectasis and emphysema, necrotic foci, leukocyte infiltration, and multiple hemorrhages.	Severe neuronal death (more than 30%), large foci of necrosis, swelling of the meninges, leukocyte infiltration, and multiple hemorrhages.

**Table 2 viruses-18-00051-t002:** Amino acid substitutions affecting the biological characteristics of HPAI H5Nx viruses. Gray shading indicates the absence of biologically significant mutations reported in the literature.

Protein	Substitution	A/74/2023	A/75/2023	Reference	Description
PB2	T271A	**T**	**T**	[23]	Increases polymerase activity in avian and mammalian cell lines
	E627K	K	**E**	[24,25]	Increases polymerase activity and virulence in mice; increases virulence in mammals; decreases polymerase activity and replication in avian cell lines and chickens
	E627V	K	E	[26]	Increases polymerase activity and virulence in mammalian cell lines
	D701N	D	D	[27]	Increases polymerase activity and replication in mammalian cell lines
	L89V	V	V	[28]	Increases polymerase activity in mammalian cell lines; increases virulence in mice
	G309D	D	D		
	T339K	K	K		
	R477G	G	G		
	I495V	V	V		
	A676T	T	T		
PB1	D3V	V	V	[29]	Increases polymerase activity and replication in mammalian and avian cells
	D622G	G	G	[30]	Increases polymerase activity in mammalian cells and virulence in mice
PA	S37A	A	A	[31]	Increases polymerase activity in mammalian cells
	S224P	**S**	**S**	[32]	Enhanced pathogenicity and viral replication in a mammalian mouse model
	N383D	D	D		
NP	N319K	K	K	[33]	Increases polymerase activity and replication in mammalian cells
NS1	P42S	S	S	[34]	Increases virulence in mice
	L103F	F	F	[35]	Increases virulence
	I106M	M	M		
	K55E	E	E	[36]	Increases replication
	K66E	E	E		
	C138F	F	F		
	V149A	A	A	[37]	Increases virulence
M1	N30D	D	D	[38]	Increases virulence in avian and mouse models
	I43M	M	M	[39]	Increases virulence
	T215A	A	A	[38]	Increases virulence in avian and mouse models

## Data Availability

Nucleotide sequences were deposited in the GISAID database (EPI_ISL_18554237, EPI_ISL_19080209, and EPI_ISL_19080210). Raw sequencing data and additional supporting data are available upon reasonable request from the corresponding author.

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
