# Peer review of "Pathobiology of Highly Pathogenic Avian Influenza A (H5N1 Clade 2.3.4.4b) Virus from Pinnipeds on Tyuleniy Island in the Sea of Okhotsk, Russia"

_viruses, 2025, doi:10.3390/v18010051_

Round 1
Reviewer 1 Report
Comments and Suggestions for Authors
Manuscript ID: viruses-3899175_Type of manuscript: Article_Title: Molecular Biological Properties of Highly Pathogenic Avian Influenza A (H5N1 Clade 2.3.4.4b) Virus from Pinnipeds, Tyuleniy Island, Sea of Okhotsk, Russia
COMMENTS TO THE AUTHORS
Brief summary
In July–August 2023, during a mass mortality of over 3,500 pinnipeds recorded on Tyuleniy Island in the Sea of Okhotsk (Russia), two highly pathogenic avian influenza (HPAI) viruses were isolated. Molecular biological properties of these (H5N1) viruses of clade 2.3.4.4b, were examined in this article.
Broad comments
In the context of the major threat to wildlife, agriculture, and public health posed by the emerging clade 2.3.4.4b of HPAI A(H5N1) virus – progressively expanding its host range and increasing the frequency of spillover into mammals − this study analyzes two HPAI A(H5N1 clade 2.3.4.4b) strains isolated during the above mass mortality from fur seal (Callorhinus ursinus) carcasses, and designated A/Northern_fur_seal/Russia_Tyuleniy/74/2023 and A/Northern_fur_seal/Russia_Tyuleniy/75/2023 strains.
With a dispatch study [citation 5 in the article under review] the Authors promptly reported in 2003 this severe outbreak caused by A(H5N1) viruses closely related to HPAI virus strains detected primarily in the Russian Far and East Japan in 2022-2023. The manuscript under review: i) updates previously reported data [5] by including in the phylogenetic analysis virus genomes circulating in 2024–2025, and by expanding the analysis of amino acid substitution reported to be associated with mammalian adaptation and increased pathogenicity in H5Nx viruses; ii) assesses in an animal model pathogenicity and pathological changes associated with A/74/2023 and A/75/2023 fur seal strains.
In my opinion, this study adds new important information to existing literature regarding the emergence of HPAI A(H5N1 clade 2.3.4.4b) viruses in marine mammals. However, I recommend major revisions aimed to improve the quality of this interesting manuscript.
Please, see “Specific comments” for details.
Specific comments:
Abstract:
- Pag. 1_27: “ mass 26 mortality of over 3,500 northern fur seals (Callorhinus ursinus) and Steller sea lions …. “
According to Ref. [5], only one sea lion was found dead. Please rephrase this sentence.
- Pag. 1_38-40: “.... The findings confirm that pinnipeds are susceptible to HPAI H5N1 clade 2.3.4.4b and highlight their potential role as reservoirs and mixing hosts for avian influenza viruses, …. “
While the results suggest that pinnipeds may play a role as mixing vessels, it is difficult to support their potential reservoir role, that would imply the HPAI H5N1 clade 2.3.4.4b virus perpetuation in these wild mammal populations in natural habitats. Is there evidence to support this, or are pinnipeds only spillover hosts able to amplify the HPAI H5N1 virus circulation? Increased surveillance will contribute to a better understanding of the epidemiological role of pinnipeds. I suggest rephrasing your above sentence.
Introduction:
In general, given the real risk posed by these HPAIVs to wildlife conservation, it would be important to include in the Introduction data on the conservation status of Callorhinus ursinus, classified as vulnerable according to the IUCN Red List. Moreover, it could be useful provide a Map showing the collocation of the sampling area or at least provide latitude and longitude coordinates of Tyuleniy Island
- Pag. 2_47-56: “ Orthomyxoviridae is a family of negative-sense RNA viruses that includes nine genera, one of which (Alphainfluenzavirus) includes avian influenza viruses (AIV), which primarily affects birds but can sometimes affect mammals including humans. Classification of a virus strain as either low pathogenic avian influenza (LPAI) or high pathogenic avian influenza (HPAI) is based on the severity of symptoms in domestic chickens. The genome consists of 8 protein-coding segments. Subtype classification is subdivided using different subtypes of the surface proteins—hemagglutinin (H) and neuraminidase (N). Currently, 16 hemagglutinin subtypes and nine neuraminidase subtypes are observed in a range of bird species. A deeper classification into clades and subclades depends on the phylogenetic parameters of hemagglutinin. “
This paragraph presents some critical issues that need to be resolved. Firstly, it completely lacks bibliographical references, which need to be added. Secondly, the classification into subtypes needs to be explained better, specifying that it is based on the combination of surface proteins, which, being 16 H and 9 N, generate a great number of possible HN combinations.
- Pag. 2_57-60 ”While HPAI has long been a threat to animal health, commercial agriculture, and food security, its risk to public health and wildlife conservation has increased dramatically. Isolation of AIVs in marine mammals, including seals, has been reported sporadically over the past 45 years. “
Pertinent bibliographical references need to be added at the end of these sentences.
- Pag. 2_73-79 ”The characterization of the subtypes associated with AIVs suggests an avian origin, which is thought to result from interspecies transmission or secondary spread from wild water birds. Reported outbreaks of AIV in pinnipeds have mainly resulted in fatal respiratory illness. Northern fur seals appear to be particularly susceptible to AIV infections, and factors such as close contact with wild birds and adaptation of virus subtypes to mammals have been suggested as factors contributing to the establishment of a potential AIV reservoir in these marine mammals.“
Please rephrase this paragraph. In the absence of bibliographical references (to be added) it is unclear whether you are referring to highly pathogenic H5 viruses or whether you are considering AIVs in general.
Regarding the role played by pinnipeds as potential reservoirs of AIVs, please, see comment to the Abstract. (Pag. 1_38-40), providing inherent references.
- Pag. 3_95 “ we have detected the HPAI H5N1 strain…”
Please amend as follows: “we have detected HPAI H5N1 strains…”
Materials and Methods:
Please, in general check carefully that this article includes the meaning of all acronyms used.
- Pag. 3_123: “…. continuous mammalian cell culture….”
I think it's better “mammalian continuous cell culture…”
- Pag. 3_131-133: 2.3. Serological Analysis
This subsection needs additional information. The serological method used as well as related references are missing.
Pag. 4_143-145: “Euthanasia and collection of internal organs for determination of virus titer and histology were allocated for three animals on days 3, 6, 10, 15, 21 after d.p.i.”
Please check the accuracy of this sentence, it is the word “after” necessary?
- Pag. 4_151: “as lg TCIDâ‚…â‚€/ml.”
Is “lg” correct?
Results:
- Pag. 5_196: “Appendix A”
Figure A1, on Page 19, consists of two different figures that should be separated from each other.
- Pag. 5_220: Thus, all H5N1 virus …..”
Perhaps you could replace “all” with “both”.
- Pag. 7_249-264: “In the lungs of mice infected ……….. (brain, A/75/2023 strain)”
Recommendation: If “lg” and “Log10“ have the same meaning, choose one of these two acronyms to be used throughout the article.
- Pag. 7_272: “…. (Figure 4a)…..”
To facilitate consultation of Figure 4, I suggest that from here on, the various photos be cited in the text (including the Results section) as follows: Figure 4_A1, Figure 4_A2, Figure 4_B1, Figure 4_B2, etc.
Moreover, in the caption of Figure 4, 3d” can be replaced with “3rd”.
- Pag. 11_389: “(Callorhinus ursinus)”
As previously done in the manuscript, use italics for the species name.
- Pag. 13_420-421: If, as I understood, the grey cells indicate in Table 1 the absence of mutations previously associated with adaptation to mammals and increased pathogenicity, why was the absence of the D701N mutation not highlighted? I also suggest specifying the meaning of the grey cells in the Table 1 caption.
- Pag. 14_441-442: I also suggest specifying the meaning of the grey cells in the Table 2 caption.
Discussion:
- Pag. 14_479-483: “….. Both H5N1 viruses caused death of the mice without prior host adaptation with evaluated 50% mouse lethal dose (MLD50) of 1.9–2.5 log10EID50/ml. H5N1 influenza viruses with MLD50 ranging from 1.6 to 5.8 log10EID50/ml can be classified as highly virulent for mice. Regardless of the MLD50 determination, ….."
I suggest replacing “MLD50” and “log10EID50/ml” with “MLD50” and “log10EID50/ml”.
Pag. 17_592-595: “Influenza may be more endemic among marine mammals than previously thought, although live virus is difficult to extract. Despite these challenges, increasing evidence suggests that marine mammals are an important wild reservoir of influenza and may facilitate mammalian adaptation to avian variants”
Regarding the role played by pinnipeds as potential reservoirs of AIVs, please, see comment to the Abstract. (Pag. 1_38-40), providing inherent references.

Author Response
Dear Reviewers and Editor,
We are thankful for all your useful comments suggestions to improve the manuscript. We have dramatically revised and modified the manuscript following these suggestions, we included additional text and figures. We completely agree with serious comments on study limitations raised by all reviewers. We have toned down our statement through the text and tried to modified all sections to make the study and interpretations clear; also we included detailed "Limitations" section. So we summarized our detailed responses in one PDF attached.
Also we would like to include additional co-authors who helped us and provided significant contribution to modified manuscript (additional analysis, writing and editing).
As all reviewers indicated to a poor English we apply for additional English editing service of Journal.
We appreciate you for your valuable feedback which helped us to improve the manuscript.
Sincerely, Authors

Reviewer 2 Report
Comments and Suggestions for Authors
Molecular biological properties of highly pathogenic avian influenza A (H5N1 clade 2.3.4.4b) virus from pinnipeds, Tyuleniy Island, Sea of Okhotsk, Russia
Authors:
Alexander Alekseev, Ivan Sobolev, Kirill Sharshov, Marina Gulyaeva, Olga Kurskaya, Maria Chisty-aeva, Alexander Ivanov, Olesia Ohlopkova, Marina Stepanyuk, Anastasiya Derko, Mariya Solomatina, Batyrgishi Mutashev, Mariya Dolgopolova, Alimurad Gadzhiev and Alexander Shestopalov.
Summary:
The authors characterised the pathobiological properties of two A(H5N1) highly pathogenic avian influenza virus (HPAIV) isolates obtained from pinnipeds in Russia. The authors examined the lethality of the isolates in chickens (IVPI) and mice (MLD50). The authors assess the pathogenicity of both isolates in mice using 10 LD50 dose and examined morbidity as measured by weight loss, mortality, virus replication in tissues and pathology. Additionally, the authors re-presented phylogenetic analysis of the A(H5N1) HPAIV isolates. Whilst the data is timely and of interest, the manuscript has been greatly impacted by the composition and poor articulation of results and discussion. The manuscript would benefit from a full revision.
General comments:
- Overall, the manuscript as it is written is confusing at times, superfluous (EG: “vivid manifestations”), some descriptions don’t make sense and conclusions contradict the data. It seems that this manuscript may have been prepared to some degree with generative AI.
- The discussion repeats results, introduces new analyses and lack focus and contextualisation of the findings with the broader literature.
- Some results have already been published in their earlier manuscript and are not referenced.
Figures:
- Figure 1 was difficult to follow. The authors may consider replacing Figure 1 with Supplementary Table B1 which was much easier to follow and understand what was happening.
- Figure 2 – This figure would benefit from showing individual data points and error bars and select appropriate statistical analysis of weight loss.
- Figure 3 – The authors should consider presenting this data another way for clarity. For example, on day 6, virus was detected in only one animal out of three (brain, A/75/2023 strain), but the bar graph without error bars does not clearly communicate this data. Lg TCID50/mL – I assume that this is Log10 TCID50/mL – please update.
- Figure 4 – The authors should highlight pathological changes in images with symbols for readers. This figure was difficult to follow – perhaps consider including the virus abbreviation in the top left corner.
- Figure 5 – The bootstrap values are pixelated and hard to read.
Specific comments:
- The authors may consider modifying their title from ‘Molecular biological properties of..’ to ‘Pathobiology of …’ to reflect the work within.
- An ethics statement for the collection of samples from living fur seals, IVPI, MLD50 and murine pathogenicity study should be included.
- Serology – the methods are lacking sufficient information and no serological results are presented.
- IVPI – Not described in methods section.
- Pathogenicity study methodology does not describe the control group – the authors can address this by moving the methodology description from the results to methods section. Key details are lacking – what was the starting point of the tissue homogenate that was titrated, EG: 10%, gm tissue/mL, etc.. The authors should specify the MLD50 for each virus tested.
- OIE is now WOAH.
- Figure 3 results – the authors report that all mice died in the pathogenicity study but then comment that the pathology indicated signs of repair for A/75/2023 (line 299). This statement contradicts the results.
- I don’t understand the value of introducing A/74-2/2023 – it seems to confuse things.
- Figure 3 - No virus replication in the lungs for A/75/2023 on D3 and other tissues of infected mice was puzzling. This should be discussed in more detail.
- The phylogenetic data is currently insufficient to support the conclusion of mammal-mammal transmission.
- Line 512 – Please note, PB2 E627K has also been identified in avian species.
- Line 54-55 ‘Currently, 16 hemagglutinin subtypes and nine neuraminidase sub-54 types are observed in a range of bird species requires a reference. Please also note/cite H19.
Line 80-86, while the authors claim that pinnipeds act as ‘mixing vessels’ for human and avian influenza virus reassortment, they do not clarify the logic behind this. In the introduction, they liken this to swine respiratory system that possess both α-2,3 and α-2,6 sialic acid receptors. However, the article the authors refer to here (Runstadler et al;, argues there is conflicting data about distribution of different sialic acid receptors in pinnipeds.
Comments on the Quality of English LanguageOverall, the manuscript as it is written is confusing at times, superfluous (EG: “vivid manifestations”), some descriptions don’t make sense and conclusions contradict the data. It seems that this manuscript may have been prepared to some degree with generative AI.
Author Response

(The authors gave the same response as above.)

Reviewer 3 Report
Comments and Suggestions for Authors
This manuscript describes the investigation of a mass mortality event among northern fur seals and Steller sea lions on Tyuleniy Island, Russia, in July-August 2023. The authors identified Highly Pathogenic Avian Influenza (HPAI) A(H5N1), clade 2.3.4.4b, as the cause. They isolated two viral strains from carcasses and characterized them through pathogenicity studies in chickens and mice, histopathological analysis of infected mouse tissues, and full-genome sequencing with phylogenetic analysis. The work aims to confirm pinniped susceptibility to this virus clade and to assess the viral properties, including the presence of mammalian-adaptive mutations. The study concludes by underscoring the potential role of marine mammals in influenza virus ecology and calls for enhanced surveillance. This manuscript, in its current form, is not suitable for Viruses. The journal typically expects a more mechanistic and molecularly deep analysis. The present study is primarily descriptive (pathology, basic virology, phylogeny) and the molecular analysis lacks functional insight. Below are my comments:
- The abstract overstates the public health implications and fails to highlight the most critical limitation: the artificial nature of the high-dose mouse infection model. The abstract concludes by highlighting the "potential role as reservoirs and mixing hosts." What specific evidence in your study demonstrates reservoir status (i.e., long-term maintenance and shedding) or mixing vessel activity (i.e., evidence of reassortment with other influenza viruses)? You have shown susceptibility and fatal outcome, which is the opposite of what is required for a reservoir host. Given that the mouse challenge was performed with an extremely high dose (10 MLDâ‚…â‚€), how do you justify extrapolating these findings to the natural transmission dynamics and disease progression in wild seals? Isn't it possible that this model primarily reflects an overwhelming, artificial infection rather than a natural pathogenic process?
- The introduction provides a general background on avian influenza but does not sufficiently focus on the specific ecological and virological knowledge gaps related to pinnipeds in the North Pacific. The rationale for the study is implied rather than explicitly built. The introduction mentions that seals can act as 'mixing vessels' but does not clarify the specific ecological and physiological conditions on Tyuleniy Island that would make this a likely scenario. What is the evidence for co-circulation of other influenza A virus subtypes in the seabird or pinniped populations on this island? You state that factors like "close contact with wild birds" contribute to AIV establishment in marine mammals. For Tyuleniy Island specifically, what quantitative data exist on the degree of contact between the dying fur seals and wild bird populations? Is this based on direct observation or inference?
- The method for virus isolation in MDCK cells is described, but no data is presented on the results of this isolation (e.g., titers, comparison to egg isolation). It is stated that "extensive cytopathogenic effects" were seen at 3 d.p.i., but this is vague and presented out of sequence in the results. Why are no quantitative data (e.g., viral titers from cell culture isolation) provided? Was the isolation in MDCK cells less efficient than in eggs? The absence of this data raises questions about the robustness of the isolation protocol used for subsequent experiments.
- The serological analysis is critically deficient. It lacks essential validation details, making the reported negative result uninterpretable and potentially meaningless. You used a single, high starting dilution (1:40). What was the rationale for choosing this dilution? Did you perform a titration series to determine the assay's endpoint? A negative result at 1:40 is not evidence of a complete lack of antibodies; it could simply mean titers were below this arbitrary cutoff. What positive control serum was used to validate the Hemagglutination Inhibition (HI) assay? How was the concentration of turkey red blood cells standardized? Without these critical controls, the statement that "there was an absence of antibodies" is not scientifically supported.
- The use of a 10 MLDâ‚…â‚€ dose is a major limitation that severely constrains the biological relevance of the findings. This approach is designed to cause rapid, severe disease and death, which may obscure more subtle and meaningful differences in viral pathogenesis and tropism. Given that the primary finding is a difference in virulence between the two strains, how can you be sure that this difference would be apparent—or even exist—under a more natural, low-dose exposure scenario? Isn't it possible that you are characterizing the response to a toxic insult rather than a replicative pathogenic advantage?
- The description of the outbreak is qualitative and lacks rigor. The case definition and methods for estimating mortality are not provided. How was the figure of "over 3,500" dead animals derived? Was this a complete census or an extrapolation? What methods were used to count animals that died in the water, as you acknowledge is a likely occurrence? You describe two stages of disease progression. Were these stages based on systematic observation of marked individuals, or are they anecdotal descriptions? How many animals were observed to progress through these stages?
- The results for weight loss and survival are presented in the main text but the critical data (Figure 1 and 2, and the table in Appendix B) are either poorly described or missing from the provided PDF. This makes independent assessment impossible.
- Appendix B Table B.1 seems to show deaths in the control group. Can you explain why control animals died or were euthanized? This raises serious concerns about the welfare of the animals and the validity of the model system. The virological data (Figure 3) show viral titers in lungs and brain. Why were other organs (e.g., spleen, kidney) not tested, especially since HPAI H5N1 is known for systemic infection? The claim that the "virus was not detected in other organs" is weak without supporting data from a broader tissue panel.
- The histopathological analysis descriptions are purely qualitative. There is no semi-quantitative scoring of the severity of lesions (e.g., severity of inflammation, percentage of lung affected), which is essential for objectively comparing the two virus strains. You state that infection with A/75/2023 was "milder" and that "processes of restoration of organ structures were observed" by day 10. What specific histological features indicate active repair (e.g., hyperplasia, fibrosis) versus simply less severe damage? The histopathology figures (Figure 4) are referenced with labels (A1, B2, etc.) that are not fully explained in the legend or text, making it difficult to match the description to the image. Furthermore, where are the images for the control group (F1, F2) to baseline normal tissue architecture?
- The phylogenetic analysis is presented as a figure (Figure 5) that is not included in the provided PDF, making its evaluation impossible. The discussion of mutations is a simple cataloguing exercise without deeper analysis of their potential functional consequences in the context of the entire viral genome. You identify PB2-E627K in one isolate. Was this mutation present as a minor variant in the other isolate, or was it truly absent? Was deep sequencing performed to rule out a mixed population?
- Table 1 lists many mutations that are described as "dominant" in circulating viruses. If these mutations are nearly ubiquitous, how can they be considered meaningful markers of mammalian adaptation in your specific isolates? Doesn't this suggest they are now the basal state for this clade and may have been selected for replication in birds?
- The discussion overinterprets the data and makes sweeping statements about public health risk and viral evolution that are not strongly supported by the experimental results. It fails to adequately acknowledge the study's limitations. You conclude that the viruses are "actively adapting to mammals." However, your data show these viruses are highly lethal in seals and mice. Lethal infection typically indicates poor adaptation, as a well-adapted virus would ideally propagate without killing its host. How do you reconcile lethality with successful adaptation? You emphasize the need for surveillance at the "wild-life–livestock–human interface." However, Tyuleniy Island is described as a remote wildlife habitat. What is the specific pathway you envision for a spillover from seals to livestock or humans in this region, and what data support the plausibility of this pathway?
- Table 1: The column headings "74/74-2" and "75" are unclear. Does "74/74-2" mean the mutation was found in both isolates from the same animal? The table needs a clearer legend explaining what each entry represents (e.g., K = Lysine present, E = Glutamic acid present).
- Figure 4 Legend: The legend is incomplete and confusing. It lists descriptions for panels (e.g., A1, B2) but does not provide a key to what Groups 1, 2, and 3 correspond to in the main text. The labels within the figure itself are not visible in the provided text.
The English could be improved to more clearly express the research.
Author Response

(The authors gave the same response as above.)

Reviewer 4 Report
Comments and Suggestions for Authors
Please, find attached.

I cannot assess the quality of English because it is not my native language.
Author Response

(The authors gave the same response as above.)

Reviewer 5 Report
Comments and Suggestions for Authors
The manuscript can be published; however, the pathology section needs to be expanded. In the Histological Analysis section, the description of the standard processing procedure can be omitted, simply by citing a reference for the standard protocol. However, this section must describe the methodology used to assess the lesions, including their type, severity, and frequency.
The results section should include a description of the pathological changes observed in the brains of mice infected with strains A/74/2023 and A/75/2023. The methodology used to assess these changes should be described in the Materials and Methods section; additionally, it is recommended to present the sequence of pathological changes in a flowchart.
In Figure 4, the histological images should be labeled to clearly indicate the lesions described in the figure caption.
Author Response

(The authors gave the same response as above.)

Round 2
Reviewer 1 Report
Comments and Suggestions for Authors
Manuscript ID: viruses-3899175_Type of manuscript: Article_Title: Molecular Biological Properties of Highly Pathogenic Avian Influenza A (H5N1 Clade 2.3.4.4b) Virus from Pinnipeds, Tyuleniy Island, Sea of Okhotsk, Russia
COMMENTS AND SUGGESTIONS FOR AUTHORS
According to the suggestions of five reviewers, the manuscript has been thoroughly revised by the authors, starting from the titles. Furthermore, in response to the reviewer 3 comments, an additional section "Limitations of the study" has been included.
In my opinion, this study adds new important information to existing literature regarding the emergence of HPAI A(H5N1 clade 2.3.4.4b) viruses in marine mammals.
- I recommend minor revisions aimed to improve the quality of this interesting manuscript.
Please, for details see “Specific comments” to the revised PDF file.
Specific comments:
Introduction:
- Pag. 2_60-61: “ Subtype classification is subdivided using different subtypes of the surface proteins ….”
The word order is rather contorted in this sentence, and it is not clear that IAVs are classified into subtypes according to the combinations of their surface proteins. I suggest to rephrase the text slightly.
- Pag. 2_66-67: “ its risk to public health and wildlife conservation has increased dramatically. [5].”
I suggest moving reference [5] from line 67 to line 68, as follows: “... over the past 45 years [5].
- Pag. 2_82-83: “The characterization of the subtypes associated with AIVs suggests an avian origin, (H7N7 [8], H3N8 [9], H4N5 [10], H3 [11], H10N7 [12] et al.) …..”
I really appreciate this information added to the manuscript, but it would be better to integrate it into the text. For example:” The characterization of H7N7 [8], H3N8 [9], H4N5 [10], H3 [11], H10N7 [12] subtypes detected in seals suggests an avian origin, which……”
- Pag. 2_83-84: “…. which is thought to result from interspecies transmission or secondary spread from wild water birds”.
What do you mean by “interspecies transmission”? Do you mean transmission between different species of mammals and/or seals? Please explain it better.
- Pag. 2-3_92-96: “Mammalian sialic acid …………….. and public health”
I recommend checking the grammar, especially in the first two sentences.
- Pag. 3_113-115: “On Tyuleniy Island we have detected the HPAI H5N1 strain belonging to clade 2.3.4.4b, isolated from northern fur seal carcasses. These are closely related to HPAI virus strains detected primarily in the Russian Far East and Japan in 2022-2023 [15].”
In the first sentence, you wrote “strain” (singular). The second sentence begins with “They are ...” (plural). Please correct and/or complete the text
Materials and Methods
- Pag.4_173: “….4 GAU……”
Please check “4 GAU”.
- Pag. 5-6: “Table 1”
Please check the Russian term in the last row of the table
Results
- Pag. 15: “Figure 5”
Please check the y-axis graphically.
Discussion
- Pag. 25_652-655: “Although we could not stste the significant differences between studied strains; these are preliminary data, they should be clarified in further studies using large samples of experimental animals, various doses of infection and methods of reverse genetics.”
Please check and rephrase this sentence.
- Pag. 27_734: “….. isolates isolated …..”
I suggest to replace “isolates isolated” with “strains isolated.
Appendix A
- Pag. 32: “Figure A.2”
I suggest to replace in the Figure A.2 caption “Its unique environments enables ….” with ”The island's unique environment enables ….. “
Appendix B
- Pag. 34: “Table B.3”
Please enter the meaning of “*” in the table footnotes.
- Pag. 35: “Table B.4”
Please enter the meaning of both “*” and “**” in the table footnotes.
Comments on the Quality of English LanguageThe English could be improved to more clearly express the research.
Author Response
Dear Reviewers and Editor,
We are thankful for all your useful comments suggestions to improve the manuscript. We have revised and modified the manuscript following these suggestions.
As all reviewers indicated to a poor English we contacted for additional English editing service of Journal.
We appreciate you for your valuable feedback which helped us to improve the manuscript.
Sincerely, Authors

Reviewer 3 Report
Comments and Suggestions for Authors
This study investigates a mass mortality event of northern fur seals on Tyuleniy Island, Russia, in the summer of 2023, linking it to the highly pathogenic avian influenza (HPAI) H5N1 virus, clade 2.3.4.4b. The researchers isolated two viral strains from seal carcasses and conducted a comprehensive analysis of their pathogenicity in chicken and mouse models, their genetic makeup, and their phylogenetic relationship to other circulating viruses. The core findings are that these seal-origin viruses are highly pathogenic in both animal models, with one strain (A/74/2023) appearing more virulent in mice, causing more severe lung and brain damage. The genetic analysis reveals these viruses are closely related to contemporary strains from the Russian Far East and Japan and carry several mutations associated with adaptation to mammals, most notably the PB2-E627K mutation in one of the isolates. The study concludes that pinnipeds are susceptible hosts and could act as "mixing vessels" for the virus, underscoring a significant threat to wildlife and public health that warrants enhanced surveillance.
General comments:
- The manuscript is difficult to follow. Descriptions of methods and results are often fragmented, repetitive, and contain contradictory information (e.g., the number of animals used, the timing of procedures). This makes it nearly impossible to critically evaluate or reproduce the experiments.
- The central narrative suggests seal-to-seal transmission and viral evolution within the seal population. However, with only two virus isolates from a single outbreak event and no contemporary viral sequences from birds on the island, these claims are highly speculative. The data presented could just as easily support two independent spillover events from an unsampled avian reservoir.
- The pathological scoring, while attempted, is not convincingly supported. The descriptions are qualitative, and the semi-quantitative scoring system lacks validation. The statistical analysis is poorly described, and it's unclear if appropriate tests were used for the type of data presented.
Specific comments:
- The abstract states the study "confirms[s] that pinnipeds are susceptible" and then later says it "represents the first documented case". Is the goal to confirm a known susceptibility or to report a novel finding? Please clarify the primary novelty of this work. Given that HPAI H5N1 clade 2.3.4.4b has already been documented in numerous pinniped species globally, what specific, unresolved question about the virus in northern fur seals does this study aim to answer?
Materials and Methods
- The authors state, "individual fur seals were carefully restrained using humane methods." What specific, standardised restraint methods were used for these wild animals to ensure minimal stress and injury? Please provide a detailed description.
- The text states "MDCK cell culture. For this purpose, MDCK cell culture." This repetition suggests a draughting error. Please provide a single, clear, and complete protocol for virus isolation in both eggs and MDCK cells, specifying the precise media composition and passage details.
- The methodology is a major weakness. You state a group of 15 mice was used per virus but then describe euthanising 3 animals on days 3, 6, 10, 15, and 21. This adds up to 15 mice, leaving no animals for the full mortality observation. Yet, you present survival data. Please provide a clear, unambiguous table or flowchart detailing the exact number of mice allocated to survival monitoring versus those allocated for scheduled organ harvest at each time point. The current description is contradictory.
- The scoring system (1-4) is semi-quantitative and subjective. Was this scoring performed by multiple pathologists who were blinded to the experimental groups? If not, how did you control for observer bias? What was the inter-observer variability?
Results
- The authors observed a two-stage disease progression in seals. Was this based on continuous monitoring of identified individuals, or is this a composite sketch from sporadic observations? How can you be sure Stage 1 lasted 6-8 hours and Stage 2 lasted 2-4 hours without intensive individual tracking?
- The authors conclude that A/74/2023 is more virulent than A/75/2023. However, the MLD50 values you report (1.9-2.5 log10 EID50/ml) are overlapping. Given the inherent variability of this assay, what statistical test confirmed that this difference is significant? Please provide the exact p-values.
- For A/75/2023 in the brain, the virus was detected in only 1 of 3 animals on day 6. How can you draw any meaningful conclusion about "varied virus titers" from a single positive sample? Wouldn't it be more accurate to state that viral neurotropism was inconsistent for this strain?
- All 11 pups were seronegative. Given the mass mortality, does this imply that the virus killed infected animals before they could seroconvert, or that the pups were simply not exposed? How does this negative result help your investigation into the outbreak dynamics?
Discussion
- The authors state, "we believe that the replacement [PB2-E627K] has probably already occurred in the seal population." You have one isolate with this mutation from one animal. How can you distinguish between this mutation arising de novo in a single infected seal during viral replication versus a stable, transmitted lineage circulating in the population? Your data from a single time point cannot establish this.
- The authors hypothesise "there was a transmission of mammal-to-mammal" because the mass death is "difficult to explain only by infection from birds." While plausible, this is not evidence. Couldn't a high density of susceptible hosts and a high force of infection from a large, unsampled wild bird outbreak equally explain the mass mortality? What specific data do you have that rules this out?
- The authors note that the PB2-E627K mutation is rare globally and in your regional cluster. If this mutation provides such a strong adaptive advantage in seals, why didn't it become fixed in the outbreak? The co-circulation of E and K variants suggests a more complex fitness landscape. What is your explanation for why this supposedly key mammalian adaptation did not take over?
Study Limitations
- The most severe limitation is the sample size of two viral isolates from a mass mortality of 3,500 animals. How can you make any definitive claims about viral evolution or transmission dynamics with such a tiny, potentially non-representative snapshot of the viral diversity present during the outbreak? The complete lack of contemporaneous avian influenza virus sequences from birds on Tyuleniy Island is a critical data gap. Without this, how can you truly reconstruct the spillover event(s) or rule out a common avian source for both of your seal viruses?
Tables and Figures
- Figure 1 / Table B.1: The figure and table are confusing and do not align well with the text. The legend uses terms like "animal live", "animal death", and "animal euthanasia", but the plot is a stacked bar chart of counts. It's visually unclear. Table B.1 is essential but poorly formatted and difficult to decipher. A clear timeline diagram would be far more effective.
- Figure 2: The weight change graph is one of the clearer elements. However, the error bars are not defined in the caption (are they SD or SEM?). This is a critical omission for assessing variability.
- Figure 3: The bar graph of virus titres is problematic. It plots mean titre, but for groups where the virus was not detected in all animals (e.g., A/75/2023 brain, day 6), what value was assigned for the animals with no detectable virus? Using a value of zero would be misleading, and omitting them changes the N. This must be explicitly stated. The caption "Virus was detected in only one animal out of three" should be on the graph itself.
- Figures 4, 5, 6, 7 and Pathological Scoring (Tables B.3, B.4): This is the weakest part of the data presentation.
- The micrograph labels (A1, A2, B1, etc.) in Figure 4 are not systematically explained. The reader must cross-reference a long and messy legend with the images.
- The pathological scoring in Tables B.3 and B.4 shows means with very small standard errors (e.g., 3.71±0.121) for a semi-quantitative score from 1 to 4. How is such high precision possible with a subjective score from a small number of samples? This raises concerns about the robustness of the scoring process.
- Figures 5 and 7, which summarise the scores, are essential but lack clarity. The statistical annotations (*, **) are explained in a note, but the actual p-values are not provided, making it impossible to judge the strength of the evidence.
- Table 2 (Amino Acid Substitutions): This table is crucial but rendered almost useless by formatting errors, placeholder text like "[H]--[H22]", and the inexplicable "Inserted Cells" text. It must be completely rebuilt for clarity and accuracy. The data within is critical to your claims and must be presented impeccably.
- Phylogenetic Figures (C.1-C.8): The resolution is too low to read. The description says they form a "Russian Far East cluster", but without clear visuals, the reader cannot evaluate this key claim.
The manuscript is replete with errors that must be fixed. A non-exhaustive list includes:
- Page 1: "Steller sea lionslion": "Steller sea lion"
- Page 2: "HSN1" repeatedly written as "HSN1".
- Page 3: "slalic acid": "sialic acid"; "bromchioles": "bronchioles".
- Page 4: "Proxon tissues": "Organ tissues"; "DMEMMEM": "DMEM".
- Page 5: "Miccin vivo": "Mice in vivo"; "insectium in each hostil": "inoculum in each nostril".
- Page 7: "IAVTue AIV": "The AIV"; "Tue 50% egg infectious dose": "The 50% egg infectious dose".
- Inconsistent use of "A/74/2023" vs. "A/74-2/2023". The decision to focus on one should be stated clearly early on.
- Numerous issues with hyphenation, spacing, and punctuation.
Author Response

(The authors gave the same response as above.)

Round 3
Reviewer 3 Report
Comments and Suggestions for Authors
Main concerns:
- The abstract states the study "represents the first documented case..." yet the introduction cites a 2016 case in grey seals. Is this claim specifically for the North Pacific regionor for clade 2.3.4.4b in pinnipeds? This needs precise clarification.
- The text refers to isolates A/74/2023, A/74-2/2023, and A/75/2023 confusingly. It's stated that A/74 and A/74-2 from the same animal are identical, so only A/74 is used. However, tables and phylogeny sometimes reference A/74-2. This creates unnecessary complexity. For absolute clarity, why not formally designate and use only two representative isolates: one from Animal 1(e.g., A/74/2023) and one from Animal 2 (A/75/2023), and remove A/74-2 from all analyses and figures?
- The use of a high intranasal dose (10 MLDâ‚…â‚€) is acknowledged as a major limitation. However, the discussion on this point is retrospective. The rationale for choosing this extreme dose, rather than one aiming to mimic a more natural exposure, is not justified in the Methods. Given that a 10 MLDâ‚…â‚€ dose is essentially a lethal overdose guaranteeing severe disease, how can the observed differences in weight lossand lesion severity between the two strains be meaningfully interpreted in the context of natural transmission potential? Doesn't this approach primarily measure the upper limit of virulence under artificial conditions?
- Testing 11 livingpups for antibodies against the virus tells us only that these specific pups had not seroconverted at the single time of sampling. It says nothing about seroprevalence in the adult population that died or in surviving adults. What was the scientific objective of this serology? Given that the outbreak caused mass mortality, wouldn't the expected result be negative? How does this data advance the study's conclusions?
- The conclusion of likely "two spillovers" and potential "seal-to-seal transmission" is intriguing but heavily speculative based on only two isolates from a single day. With only two viral sequences from the entire mortality event, how can the authors robustly distinguish between (a) two independent bird-to-seal spillovers, (b) one spillover followed by rapid seal-to-seal spread with minimal mutation, or (c) one spillover with some initial diversity in the bird population? Isn't this conclusion underpowered?
- The PB2-E627K mutation is found in one isolate (A/74/2023) and framed as a key adaptation. However, the data shows A/75/2023 (without E627K) was also highly lethal in mice and caused significant disease. The authors note E627K is rare even in mammalian isolates. If A/75/2023 (lacking E627K) caused mass seal mortality and severe disease in mice, what is the functional and epidemiological evidencethat E627K is a critical driver of this outbreak, rather than a coincidental mutation in one infected animal? Does your data suggest the underlying viral "backbone" (with other common mutations like L89V, G309D, etc.) is sufficient for high pathogenicity in seals?
- The semi-quantitative scoring system (1-4) is described, but the statistical analysis of these ordinal scores is not detailed. Using means and standard errors (SEM) for ordinal (non-parametric) data is questionable. The labels in Table B.3/B.4 (e.g., 2.02+0.131*) are cryptic. What specific statistical test was used to compare the pathology scores (e.g., Mann-Whitney U)? What do the asterisks (*) and symbols (°) in the tables denote? The statistical approach must be transparent and appropriate for the data type.
- Figure 3 and the text report virus detected in the brain of only 1 of 3 mice for A/75/2023 at day 6. This is presented without comment on the implications. Does this sporadic detection suggest a lower neurotropism for A/75/2023, or is it simply a limitation of sampling? How does this fit with the histopathological evidence of brain lesions in all groups?
Tables and Figures Review:
- Figure 4", "Figure 58"). Captions are incomplete, and labels within figures (e.g., A1, B2, ✔) are not adequately explained in the legend.
- Table 1/2 (pp5-6, 22-23): Table 1 mixes descriptions for lungs and brain confusingly. Table 2 (amino acid substitutions) has formatting errors, grey shading missing, and references listed as "[H]--[H22]". The information is vital but requires complete reformatting and proofreading.
- Figure 2 (Weight change): The y-axis is labelled "Weight, %" twice. but the graph itself is a placeholder.
- Figure 3 (Viral titers): The caption is confusing ("Virus was detected in only one animal out of three..."). The graph layout is unclear. Which group is "control" on the brain panel? This needs complete redesign for clarity.
English and Typo Errors:
- P1: "Steller sea lionslion", "A/Northern_fur_seal/Russia_Tyuleniy/74/2023" (inconsistent formatting).
- P2: "more vessel for the evolution", "Keywords:highly pathogenic" (missing space).
- P4: "Proxon tissues" (presumably "Organ tissues"), "Monolayer changes-trypsin" (gibberish).
- P5: "Laboratory Miccin vivo", "10 MLD50 in practice inoculations (25 µl of insectium in each hostil: 10 MLD50)" – this sentence is garbled.
- Throughout: "Vinues", "Viraves", "dendrograms were generated and visualized with MEGAS" (likely MEGA).
The English could be improved to more clearly express the research.
Author Response
Dear Reviewer,
Thank you for your thorough analysis of our manuscript. The English language has been corrected earlier by the Journal's English editing service. Below we present our responses to the comments. In order for you to have the opportunity to see the changes in the text and distinguish them from the changes of the previous review round, all changes (text adjustments) are highlighted blue. We hope that we have responded to all your comments! Sincerely, the Authors
